# WHEN IS ADVERSARIAL ROBUSTNESS TRANSFERABLE?

## ABSTRACT

Knowledge transfer is an effective tool for learning, especially when labeled data is scarce or when training from scratch is prohibitively costly. The overwhelming majority of transfer learning literature is focused on obtaining *accurate* models, neglecting the issue of adversarial robustness. Yet, robustness is essential, particularly when transferring to safety-critical domains. We analyze and compare how different training procedures on the source domain and different fine-tuning strategies on the target domain affect robustness. More precisely, we study 10 training schemes for source models and 3 for target models, including normal, adversarial, contrastive and Lipschitz constrained variants. We quantify model robustness via randomized smoothing and adversarial attacks. Our results show that improving model robustness on the source domain increases robustness on the target domain. Target retraining has a minor influence on target model robustness. These results indicate that model robustness is preserved during target retraining and transfered from the source domain to the target domain.

## 1 INTRODUCTION

Since their proposal, neural networks are constantly evolving as they are being adapted for many diverse tasks. They have a tendency to become more complex and larger, since e.g. overparamatrization has proven to be highly beneficial. Training such large and complex neural networks usually requires a huge amount of (labeled) high-quality data. Since this amount of data is not available in all domains, transfer learning was proposed. The idea is to transfer the knowledge of a trained model from the so called source domain to a similar, related task in a target domain for which only a small amount of data exists. Usually, the transfer is considered successful if the model achieves high accuracy on the target domain. However, accuracy is not the only desired property of neural networks. Adversarial robustness is often equally important, especially in safety-critical domains. Some techniques applied in transfer learning (Shafahi et al., 2020; Chen et al., 2021) claim that they improve robustness of transfer learning. However, there is no study that directly compares these techniques to standard methods for improving robustness such as adversarial training or training with a (local) Lipschitz constant. We fill this gap by answering the following questions:

1. Which training procedure results in the most robust source models?
2. Is robustness preserved during target retraining?
3. Does robust retraining on the target domain improve robustness?
4. Which training/target retraining provides models that are robust against distribution shifts?
5. Does transferability correlate with model robustness?

To answer these questions, we use a popular transfer learning framework consisting of two parts (see Figure 1): a feature extractor $f$ which extracts representations from the inputs and is trained on the source domain and a classifier $h$ which maps extracted representations to predictions and is retrained on the target domain. We investigate and compare how different training procedures and target retraining techniques affect performance and robustness of this model. More specifically, we compare 10 training procedures that can be grouped in three categories. Category one consists of training methods that aim at achieving robustness by changing inputs, i.e. (1) training on clean inputs (ce), (2) randomly perturbed inputs (ceN) and (3) adversarially perturbed inputs (ceA), (4) supervised

contrastive learning (con) (Khosla et al., 2020), (5) supervised contrastive learning based on (5) randomly perturbed inputs (conN) and (6) adversarially perturbed inputs (conA). The second category of training approaches consists of methods that change the latent space of the model to achieve robustness, i.e. (7) latent adversarial training (feA) (Singh et al., 2019), (8) adversarial representation loss minimization (feD) (Chen et al., 2021) and (9) a combination of supervised contrastive learning and adversarial representation loss minimization (conF). Our third category of methods uses constraints on the whole model to improve robustness. These constraints are realized by (10) training with a local Lipschitz constant (llc) (Huang et al., 2021). In order to analyze how target retraining affects model robustness we compare target retraining on (a) clean ($R_{ce}$), (b) randomly perturbed ($R_{ceN}$) and (c) adversarially perturbed inputs ($R_{ceA}$).

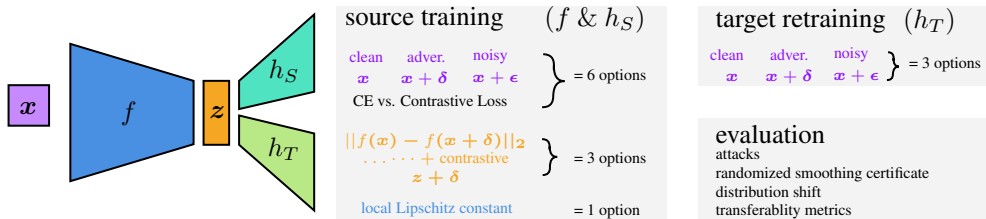

Figure 1: Transfer learning framework consisting of a feature extractor $f$, classifier $h_S$ on the source domain and $h_T$ on the target domain. For input $x$, $f(x) = z$ provides the features and $h_S(z) = h_S(f(x))$ or $h_T(z) = h_T(f(x))$ the output. Source training procedures is grouped in methods that change inputs (ce, ceN, ceA, con, conN, conA), methods that change the latent space (feD, feA, conF) and methods that constrain the whole model (llc).

To provide a more complete picture of robustness we consider robustness certification, performance against a variety of attacks, and performance under distribution shift. Namely, we employ (i) randomized smoothing based certification and (ii) Fast gradient sign method (FGSM), (iii) Project Gradient Descent (PGD) and (iv) DeepFool (DF) attacks on the source domain and the target domain. In terms of distribution shift, we determine source and target accuracy under different shifts based on random noise, changes of contrast, and Gaussian Blur shift. Next, we investigate whether there is a correlation between transferability and model robustness. We compute a transferability metric and analyze it together with model robustness and zero-shot performance. For transferability quantification we use the H-score, proposed by (Bao et al., 2019) to quantify the usability of representations learned on a source domain for learning a target task.

This battery of robustness tests can tell us when is adversarial robustness transferable. As we will show in Section 4, target models inherit robustness from the source models while target retraining has a minor impact. Our findings suggest that model robustness is transferable when source models are trained based on a procedure that enhances model robustness without being too focused on data-specific adversarial examples.

## 2 BACKGROUND AND RELATED WORK

Robustness is widely studied for standard tasks such as classification and regression, but there are few works that analyze how robustness properties can be transferred from the source to the target domain. There are different aspects of robustness. One aspect is the vulnerability to adversarial examples (Szegedy et al., 2014) – small input perturbations that are carefully-crafted to manipulate the predictions of a model (e.g. cause misclassification). Finding attacks for a given model has been widely studied for different threat models. These attacks can be used to compute an upper bound on the accuracy under adversarial perturbations. However, this bound can be loose since properly evaluating adversarial robustness is challenging. While a model may be robust against a particular attack there is usually no guarantee that it will not fail against a better and stronger attack. The lesson that seemingly robust models can be broken has been learned more than once (Carlini & Wagner, 2017; Athalye et al., 2018; Tramèr et al., 2020).

A complementary strategy to evaluate adversarial robustness is via verification/certification. Robustness certificates provide guarantees that the prediction of model will not change for the specified

perturbation set. Since certificates are NP-hard to compute in general, verification uses tractable (but sound) relaxations to provide a lower bound on adversarial robutness. Verification methods can be grouped in different categories, such as methods based on smoothing (Cohen et al., 2019), Lipschitz bounds (Fazlyab et al., 2019), interval bound methods (Mirman et al., 2018), and optimization (Wong & Kolter, 2018). Randomized smoothing (Cohen et al., 2019) is widely used due to its generality – it treats the model as a block box and only requires accesses to model inputs and outputs. Since most of the other techniques do not scale to the models typically used in transfer learning and/or are only applicable to specific families of models we adopt randomized smoothing in our evaluations.

Another aspect is robustness to distribution shift, which arise from (natural) variations in the data such as noise, changes in contrast, lighting conditions, or object composition. For example, Hendrycks & Dietterich (2019) investigate robustness using synthetic distribution shifts by introducing noise (Gaussian, shot noise), blurring, simulated weather conditions, contrast change, and corruptions from compression. To investigate this aspect we also adopt a similar procedure focusing on three types of representative shifts (noise, blurring and contrast).

The literature on robustness of transfer learning is scarce, especially relative to standard supervised learning. The few existing studies are disconnected, the findings are not comparable with each other, and even lead to contradictory conclusions. While Salman et al. (2020) shows that robust training improves the accuracy on the unperturbed target domain data, Shafahi et al. (2020) shows the opposite – adversarial training increases robustness but decreases accuracy. This is one motivation for our study – we intend to provide a fair and comparable evaluation of the most promising techniques aiming at improving robustness.

One of the strongest empirical defences is adversarial training (Goodfellow et al., 2015; Madry et al., 2018)[1] Until now, transfer learning techniques mainly use adversarial training to obtain feature representations that generalize better. Salman et al. (2020) and Utrera et al. (2021) show that adversarially trained/robust models indeed transfer better than their standard-trained counterparts, especially if the target domain has limited data. However, the primary goal of these works is to improve accuracy on unperturbed (rather than adversarial) target data. Similarly, Engstrom et al. (2019), Ilyas et al. (2019), and Allen-Zhu & Li (2021) show that adversarial training improves feature learning and results in representations that are more aligned with humans.

Goldblum et al. (2020) and (Vaishnavi et al., 2022) investigate whether robustness can transfer from an adversarially trained teacher to a student within the *same domain* via knowledge distillation. The goal for the student is to match the model output (Goldblum et al., 2020) or match the learned representations (Vaishnavi et al., 2022). In contrast to that, we investigate robustness transferability across domains. Chan et al. (2020) argues that matching input gradients is important for robustness transfer. Yamada & Otani (2022) investigate whether robustness transfers to downstream tasks such as object detection and semantic segmentation. They find that in the fixed-feature setting robustness is partially preserved and opposed to previous findings, show that an adversarial prior does not help for robustness transfer. Nern & Sharma (2022) investigate transfers from pre-trained models and theoretically shows that downstream robustness is bounded by the robustness of the underlying representation (irrespective of the pre-training protocol).

Finally, most closely related to our work is the study by Shafahi et al. (2020) showing that adversarial training of the feature extractor coupled with a one-layer-classifier improves robustness on the target domain. With a similar goal, Chen et al. (2021) proposes an adversarial training procedure that minimizes the distance between adversarial and unperturbed representations (i.e. the output of the feature extractor $f$) and proposes to use a classifier with a fixed Lipschitz constant. In all above works the robustness (when evaluated) is considered only w.r.t. a small set of attacks. To provide a more complete picture, we additionally consider verification and robustness to distribution shift.

## 3  MODEL, TRAINING PROCEDURES & TARGET RETRAINING

A simple but popular transfer learning framework, that we use for this work consists of two parts: a so called feature extractor $f$ and a classifier $h$ (see Figure 1). The prediction for input $\boldsymbol{x}$ is obtained as $y = h(f(\boldsymbol{x}))$. The feature extractor is trained on the source domain and is then frozen, i.e. not

---

[1]The details are important since adversarial training against a weak attack produces a weak defense.

changed during target retraining, while the classifier is retrained on the target domain. The idea of this model is, that in related tasks similar features are important and thus the feature extractor can be transferred from source domain to the target domain without adaptions, while the classifier maps extracted representations/features to classes and must be adapted to the target domain.

We compare the following 10 training schemes, to determine the best way of obtaining robust source models and preserving robustness during transfer to the target domain (details see Appendix A.1).

**1. Standard supervised learning (ce).** As baseline, the whole model $h \circ f$ is trained on clean input data using the cross-entropy (CE) as loss function.

**2. Randomly perturbed inputs (ceN).** We train the whole model $h \circ f$ on randomly perturbed inputs. The noisy inputs are obtained by randomly sampling the perturbation $\epsilon$ from a Gaussian distribution $\mathcal{N}(\mathbf{0}, \delta^2 \boldsymbol{I})$ and adding it to the clean input $\boldsymbol{x}$.

**3. Adversarially perturbed inputs (ceA).** During adversarial training, an attack is used to compute an adversarial perturbation $\boldsymbol{\delta}$ for each input $\mathbf{x}$. The model $h \circ f$ is trained on the perturbed inputs $\mathbf{x} + \boldsymbol{\delta}$. We use a 10-step project gradient descent (PGD) attack to obtain $\boldsymbol{\delta}$ during training.

**4. Minimizing adversarial feature loss (feD).** (Chen et al., 2021) proposed a method explicitly aimed at improving robustness of transfer learning models. This approach is based on a loss function that linearly combines the cross entropy loss $L_{\mathrm{CE}}$ with a so called representation distance loss $L_{\mathrm{R}} = ||f(\boldsymbol{x}_{\mathrm{adv}}) - f(\boldsymbol{x})||_2$, where $||.||_2$ is the $L_2$-norm. The $L_{\mathrm{R}}$ loss minimizes the distance between clean and adversarially perturbed inputs in *representation* space. The adversarial inputs $\boldsymbol{x}_{\mathrm{adv}}$ are obtained with the attack described in paragraph 3 (ceA). The final loss $L = L_{\mathrm{CE}} + \frac{\lambda}{D_R} L_{\mathrm{R}}$ is a liner combination of the two, where $\lambda \in [0; 1]$ is a hyper-parameter and $D_R$ is the dimensionality of the representation space.

**5. Latent perturbations (feA).** The previous training procedures focused on adversarial examples in the input space and/or the representation space. However, non-robustness can be caused by any layer of the neural network that maps close points far from each other. To address this issue latent adversarial training was proposed (Singh et al., 2019). The authors choose a layer $l$ and split the neural network $n$ at that layer into two parts $n_1$ and $n_2$ such that $n = n_2 \circ n_1$. Then they use the fast gradient sign method (FGSM) on the sub-network $n_2$, which results in and adversarial example in the input space of $n_2$, i.e. the latent space of the whole neural network $n$. For training, the authors additional compute an adversarial example in the input space of $n$ and combine the gradients of adversarial input and latent adversarial example to update the neural network parameters. (Singh et al., 2019) proposes to use this training procedure for fine-tuning after the training. Since standard adversarial training is not used after (but during) training, we modify their approach to ensure a fair comparison. First, we use latent adversarial training from the beginning of the training. Second, for each training step we randomly chose a latent layer $l$ for splitting the network and computing latent adversarial examples. Finally, we extended the method such that latent adversarial examples can be computed by any attack strategy. To ensure comparability we compute latent adversarial examples and adversarial inputs by using 10-step PGD attacks.

**6. Local Lipschitz constant (llc).** Bounding the Lipschitz constant of a neural network is known to improve model robustness and can even be used to get guarantees. Since bounds on global Lipschitz constants are often loose and might lead to over-regularization, we train a neural network based on an upper bound on a trainable *local* Lipschitz constant as proposed by (Huang et al., 2021). The local Lipschitz constant is obtained by taking interactions between weight matrices and activation functions into account. It can be proven that the obtained bound is tighter than the global Lipschitz constant. The details of this training approach are explained in (Huang et al., 2021).

**7. Supervised contrastive learning (con).** Neural networks trained with fully-supervised contrastive learning (Khosla et al., 2020) consist of the same two parts as our transfer learning model: a feature extractor $f$ that computes a representation $f(\boldsymbol{x})$ for each input $\boldsymbol{x}$ and a classifier $h$ that maps the representation to the output space $h(f(\boldsymbol{x}))$. The idea of contrastive learning is to compute representations for a batch of samples and train the feature extractor by pulling representations that correspond to the same class (positive samples) together and pushing representations of different classes (negative samples) apart from each other. To ensure positive samples, on each input in a batch $B$ we perform two different realizations of random data augmentations $\mathrm{aug}(.)$, such as random crops, random grey-scale changes, etc., which results in two new training batches: $B_1 = [\mathrm{aug}(\boldsymbol{x}) \, \forall \boldsymbol{x} \in B]$

$B_2 = [\text{aug}(\boldsymbol{x}) \; \forall \boldsymbol{x} \in B]$. We train on $B_1 \cup B_2$. We include this training procedure since it is directly based on a desirable property of the representations: Inputs of the same class should result in close representations. Enforcing this might affect source or target model robustness.

In each training epoch, we alternatingly update the parameters of the feature encoder $f$ and the classifier $h$. First, the feature encoder is updated based on the representation computed for the inputs using the contrastive loss (that minimizes the distance between positive samples and maximizes the distance between negative samples). Second, the representations are propagated through the classifier and the classifier is updated using cross-entropy loss.

**8. Supervised contrastive learning on randomly perturbed inputs (conN).** This training procedure is exactly like supervised contrastive learning, except for the data augmentation of each batch. We use a clean version of the batch and a version that contains randomly perturbation samples: $B_1 = [\text{aug}(\boldsymbol{x}) \; \forall \boldsymbol{x} \in B]$, and $B_2 = [\boldsymbol{x} + \boldsymbol{\epsilon} \; \forall \boldsymbol{x} \in B_1]$. The perturbation $\boldsymbol{\epsilon}$ is obtained by sampling from a Gaussian distribution $\mathcal{N}(\boldsymbol{0}, \delta^2 \boldsymbol{I})$.

**9. Supervised contrastive learning on adversarially perturbed inputs (conA).** This training procedure is again exactly like supervised contrastive learning, except for the data augmentation. We use a clean version of the batch and an adversarially perturbed version: $B_1 = [\text{aug}(\boldsymbol{x}) \; \forall \boldsymbol{x} \in B]$ $B_2 = [\boldsymbol{x} + \boldsymbol{\delta} \; \forall \boldsymbol{x} \in B_1]$. The adversarial perturbation $\boldsymbol{\delta}$ is obtained by computing a 10-step PGD attack on each input $\boldsymbol{x} \in B_1$.

**10. Fine-tuned contrastive learning (conF).** We propose to combine supervised contrastive learning as described in paragraph 7 with fine-tuning based on minimizing the adversarial feature loss. Since the standard contrastive learning operates in the representation space, but does not see any adversarial examples during training, we propose to add fine-tuning on the source dataset to increase robustness. After training, we retrain the whole model (on the source dataset) by minimizing the feature loss (see paragraph 4).

**Target retraining.** Since the feature extractor $f$ is fixed during retraining on the target domain, techniques such as constrastive learning, minimizing adversarial feature loss or latent adversarial training are not applicable to adapt the classifier to the target domain. We use and compare three different retraining procedures for the classifier: standard supervised leaning ($R_{ce}$, see paragraph 1), training on randomly perturbed inputs ($R_{ceN}$, see paragraph 2) and training on adversarially perturbed inputs ($R_{ceA}$, paragraph 3).

## 4 When is Adversarial Robustness Transferable?

### 4.1 Which training procedure results in the most robust source models?

Our goal is to analyze if and how robustness can be preserved during transfer from the source domain to the target domain. Thus, we first need to obtain robust source models. To achieve this we train models based on the 10 procedures discussed in Section 3. All models are trained on three different datasets, i.e. SVHN, EMNIST and CIFAR10. Details on the experimental set up can be found in the appendix A.1.

We evaluate model robustness in two complementary ways by using attacks and verification/certification. More specifically we use attacks of different strength, i.e. fast gradient sign method (FGSM) as a weak attack, and projected gradient descent (PGD) and DeepFool (DF) as strong attacks. For certification we use randomized smoothing, since this verification technique treats the model as black box and thus is applicable on all model architectures and activation functions. Figure 2 illustrates the results of our robustness evaluation. The exact numbers and the verifiable radius can be found in Table 1 (Appendix A.2).

First, we observe that all training procedures, except ceN and llc, result in comparable and high base accuracy ($A_{\text{base}}$ rose/gray). Training on randomly perturbed inputs (ceN) can reduce the base accuracy by $0 - 5\%$ and training with a local Lipschitz constant (llc) by $1 - 20\%$ depending on the dataset.

The verifiable accuracy ($A_{\text{cert.}}$), i.e. the portion of points for which randomized smoothing could certify the prediction, varies between the different training procedures. Not surprisingly, models trained on randomly perturbed inputs (ceN and conN) have the highest verifiable accuracy. Using

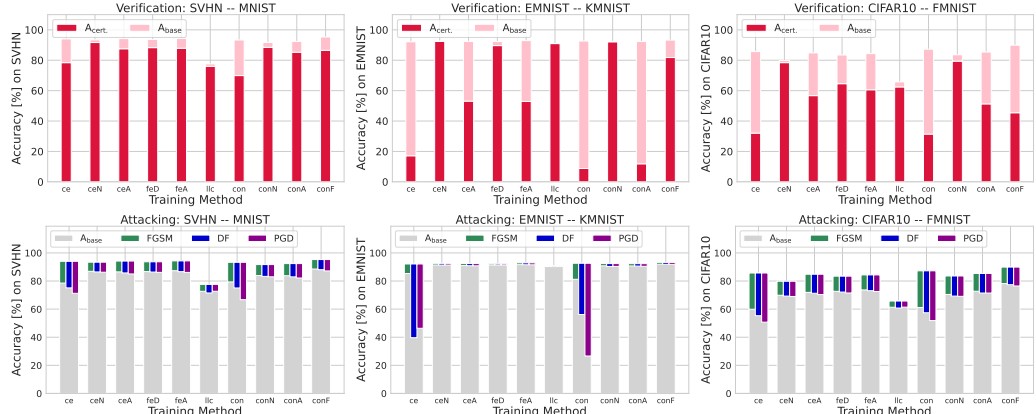

Figure 2: Accuracy on the clean test set ($A_{base}$, rose or gray), verifiable accuracy ($A_{cert.}$, red) and accuracy decrease under FGSM (green), DeepFool (DF, blue) and PGD attacks (purple) of source models trained on SVHN , EMNIST and CIFAR10.

robust training that includes attacks (ceA, feD, feA, conA, conF) or a local Lipschitz constant (llc) result results in models with medium to high verifiable accuracy and using training without perturbations (ce, con) results in the least robust models according to randomized smoothing. Our attack analysis (Figure 2) shows that using robust training (ceN, ceA, feD, feA, llc, conN, conA, conF) results in models that are significantly more robust than normally trained models (ce, con).

Thus, to obtain high accuracy and high robustness on the source dataset, we recommended to use a robust training procedure. The best performing training procedure w.r.t. to base accuracy, verifiable accuracy and robustness against adversarial attacks is conF, i.e. supervised contrastive learning followed by a fine tuning step that uses adversarial attacks and a loss function that minimizes the distance between clean and adversarial representations in the feature space (see Table 1).

## 4.2 IS ROBUSTNESS PRESERVED DURING TARGET RETRAINING?

In the previous section we analyzed how to get (the most) robust source models. Now we analyze if robustness is preserved during transfer from the source domain to the target domain during target retraining. Since we focus on inherited robustness properties in this chapter, we do the target retraining on clean inputs ($R_{ce}$). Figure 3 shows the target robustness versus the source robustness of the models. Robustness is quantified by determining the verifiable accuracy using randomized smoothing (first row) and the accuracy under the most successful (FGSM, PGD, DeepFool) attack, i.e. the attack which results in the largest accuracy decrease (second row).

If source robustness would be hundred percent preserved during transfer, the measurements would fall on the line $y = x$. However, we observe a more complex and dataset dependent correlation between target robustness and source robustness. First, models with low source accuracy result in models with low target accuracy such as ce. Models with high source robustness have different capabilities of preserving this robustness during target retraining. The amount of robustness that can be preserved depends on the source training procedure and the transfer learning task (i.e. source and target domain).

On SVHN – MNIST we observe a clear target robustness ranking of training procedures that result in similarly robust source models. Using a local Lipschitz constant (llc), contrastive learning on clean (con), randomly (conN) or adversarial perturbed inputs (conA) results in robust target models which are even more robust than the source models (measurements above $y = x$ line). On EMNIST – KMNIST the highest target robustness is observed for llc and models trained with a loss function that minimizes representation distances corresponding to clean and adversarial inputs (feD), but target models are less robust than source models. If models are transferred from CIFAR10 to FMNIST, the highest target robustness transfer is achieved by llc. The other approaches are able to preserve (most of) the robustness during transfer.

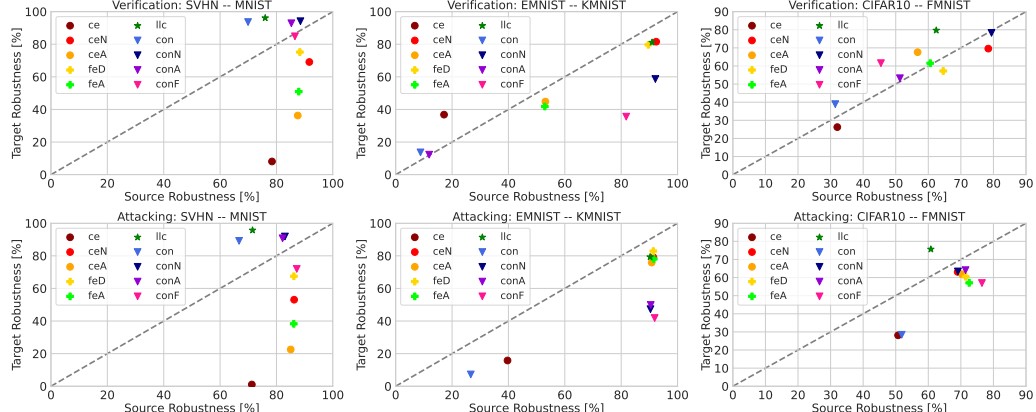

Figure 3: Target vs. source robustness quantified by verification or the most successful (FGSM, PGD, DeepFool) attack. Labels refer to the source training, target retraining is done on clean inputs.

Thus, even though they are not the most robust methods on the source domain, the most robust methods on the target domain are the contrastive learning techniques and llc which uses a local Lipschitz constant. These training procedure achieve robustness without being too focused on adversarial examples. While llc aims at mapping close input to close outputs, the contrastive approaches minimize representation distances of inputs corresponding to the same class. Thus, both approaches achieve robustness by more general concepts that computing worst case perturbations/adversarial examples which might be too data-specific to ensure that robustness is passed onto the target models.

## 4.3 DOES ROBUST RETRAINING ON THE TARGET DOMAIN IMPROVE ROBUSTNESS?

In the previous section we show that the training procedure significantly affect model robustness on the target domain. In this section we analyze the influence of target retraining on target robustness. To this end we compare 3 different target retraining procedures, i.e. target retraining on clean inputs ($R_{ce}$), on randomly perturbed inputs ($R_{ceN}$) and on adversarial perturbed inputs ($R_{ceA}$). Figure 4 compares accuracy, verifiable accuracy and accuracy decrease under attacks of these three target retraining schemes. Detailed numbers corresponding to these plots (and further plots) can be found in Table 2, 3, 4 and Figure 8 in the appendix A.2.

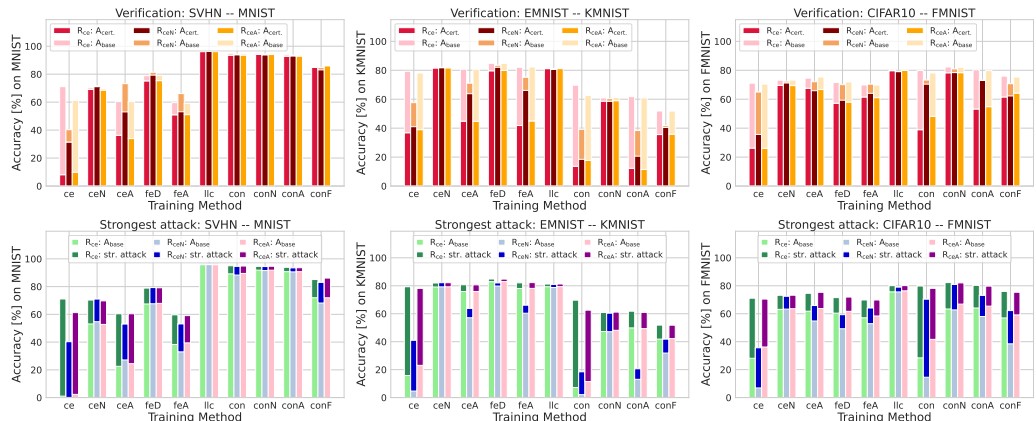

Figure 4: Accuracy on the clean test set ($A_{base}$, bright colors), verifiable accuracy ($A_{cert.}$, dark colors) and accuracy decrease of the strongest (FGSM, PGD, DeepFool) attack of target retraining on clean ($R_{ce}$), randomly perturbed ($R_{ceN}$) and adversarially perturbed ($R_{ceA}$) inputs.

Comparing the three target retraining schemes, training on randomly perturbed inputs inputs ($R_{ceN}$) results in higher verifiable accuracy (first row, dark brown bars) but decreases the accuracy on the

clean dataset. Our attack analysis shows small differences in model robustness for the three target retraining schemes. Both analyses clearly illustrate that the training procedure on the source domain has a major influence on target model robustness, while the target retraining has a minor effect on it. Thus, in order to obtain robust target models we require robust training of the source model.

### 4.4 WHICH TRAINING/TARGET RETRAINING PROVIDES MODELS THAT ARE ROBUST AGAINST DISTRIBUTION SHIFTS?

In the previous sections, we quantify and analyze adversarial robustness, i.e. robustness against small input perturbations that aim at fooling the model to make a wrong prediction. Another type of perturbations that occur in real-life transfer learning are distributions shifts. Distribution shifts are changes of the dataset such as random noise, changes in contrast or Gaussian blur. We analyze robustness of our source models (see Figure 5 and Table 5) and target models (see Figure 5, Table 6, 7 and 8) against these data shifts.

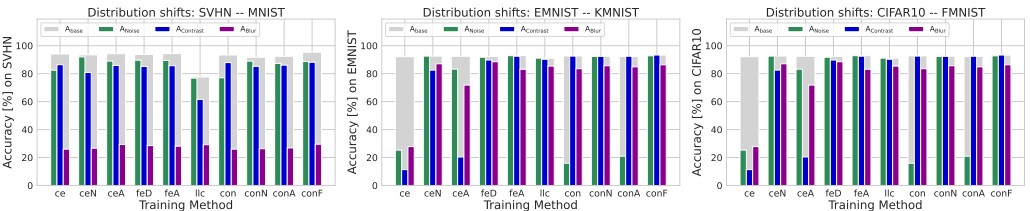

Figure 5: Accuracy on the clean test set ($A_{base}$, gray) and accuracy under distribution shifts based on random noise, changes of the contrast and Gaussian blur of source models.

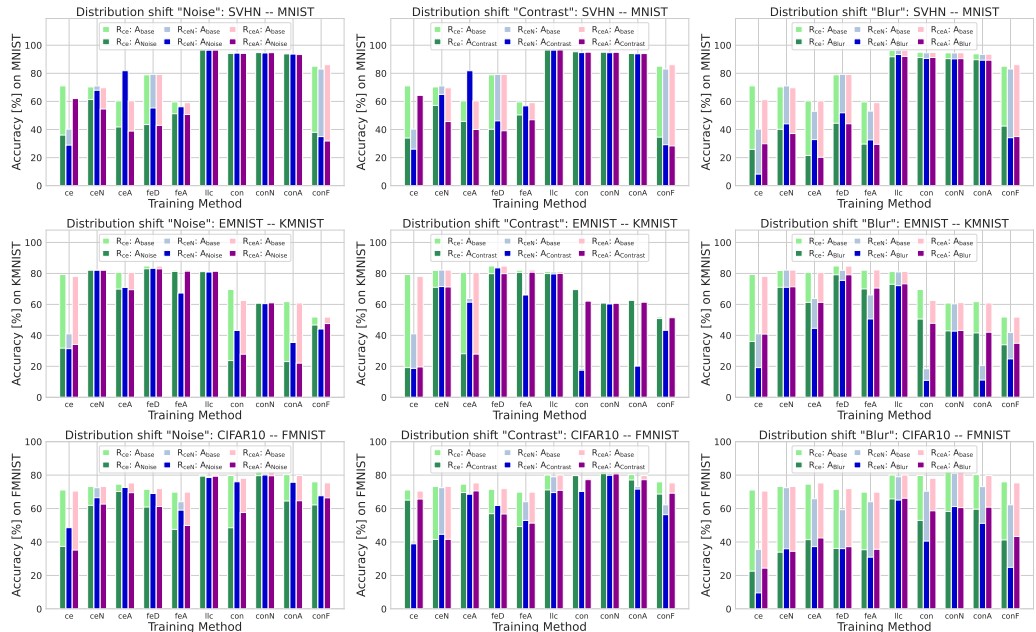

Figure 6: Accuracy on the clean test set ($A_{base}$, bright colors) and accuracy under distribution shifts based on random noise, changes of the contrast and Gaussian blur of target models retrained on clean ($R_{ce}$), randomly ($R_{ceN}$) or adversarial perturbed ($R_{ceA}$) inputs.

Robustness of source models against distribution shifts depends on the dataset and the shift. The normally trained model (ce) is the least robust model. Models trained on randomly perturbed inputs (ceN, conN) are robust to noise shifts, while contrastive learning (con, conN, conA, conF) result in the most robust models w.r.t. changes of the contrast (see Table 5). Robustness against Gaussian blur shifts can be increased by robust training or contrastive learning.

Target model robustness against distribution shifts mainly depends on the training procedure, while target retraining has a minor effect. Adversarial training and contrastive learning improve robustness against distribution shifts. On the SVHN – MNIST task training with a local Lipschitz constant (llc) and contrastive learning techniques without fine-tuning (con, conN, conA) result in the most robust target models against distribution shifts caused by noise, contrast changes or Gaussian blurring. On EMNIST – KMNIST adversarial training with a loss that minimizes distance between representations corresponding to clean and adversarial perturbed inputs (feD) yields the most robust models against all three analyzed distribution shifts. On CIFAR10 – FMNIST contrastive learning on randomly perturbed inputs (conN) and training with a local Lipschitz constant result in the strongest performing models against noise, contrast change and Gaussian blur data distribution shifts. Thus, models that are robust against adversarial attacks are also more robust against distribution shifts compared to non-robust models.

### 4.5 Does transferability correlate with model robustness?

One key requirement of transfer learning models is transferability, i.e. the potential of a model to benefit the target task. We analyze if there is a correlation between robustness and transferability of our source models (see Figure 7, Figure 13, Table 9 and 10 in the Appendix). In order to quantify transferability we use the H-score as proposed by Bao et al. (2019) and determine the zero-shot performance of the source models. The zero-shot performance is the accuracy a source model achieves on the target dataset before any target retraining.

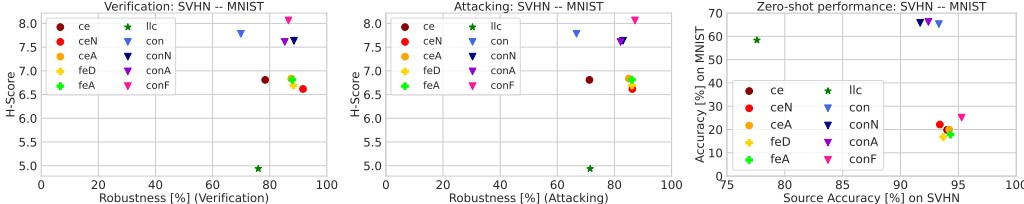

Figure 7: Transferability (H-Score) versus robustness of the source models quantified as verifiable accuracy or accuracy under the strongest attack and zero-shot performance on SVHN – MNIST.

The absolute value of the H-score depends on the dataset. Considering the ranking, the contrastive learning approaches (con, conN, conA, conF) have the highest H-scores on all three transfer learning tasks. A reason might be that these methods train the feature extractor by pulling together positive anchors (representations corresponding to the same class) and pushing apart negative anchors (representations corresponding to different classes) in the feature space. Since con performs similar to the more robust contrastive approaches we could not find a correlation between robustness and transferability. On the SVHN – MNIST task source and target dataset share all ten classes and the transferability estimations are are consistent with the zero-shot performance. The contrastive approaches have a high zero-shot accuracy of up to 66 %. On EMNIST – KMNIST and CIFAR10 – FMNIST the source dataset and the target dataset contain different classes so zero-shot accuracy is similarly low for all methods as expected (see Figure 13 in the Appendix). Thus, if the source dataset and target dataset contain the same classes the supervised contrastive learning schemes achieve the highest transferability, zero-shot accuracy and can preserve robustness during transfer.

## 5 Conclusion

This work analyzes how different training procedures on the source domain and fine-tuning strategies on the target domain affect model robustness. We show that the training procedure on the source domain has a major effect on target model robustness while target retraining has a minor effect. Our results indicate that contrastive learning and training with a local Lipschitz constant best preserve robustness during target retraining. Furthermore, robustness to adversarial attacks also provides robustness against distribution shifts. Transferability and zero-shot performance depend on the relatedness between the source and the target domain and on the source training process. The highest transferability and zero-shot performance is achieved by contrastive learning approaches, which are also among the strongest ones in preserving robustness during transfer.

## 6 REPRODUCIBILITY STATEMENT

Detailed information about the datasets, the transfer learning tasks, models, attacks and randomized smoothing is provided in the subsections of section A.1

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

# A APPENDIX

## A.1 DETAILS OF THE EXPERIMENTAL SETUP

**Models.** All models have a similar base architecture, proposed by (Huang et al., 2021). The encoder consists of 6 convolutional layers (kernel sizes: 3, 3, 4, 3, 3, 4), each one followed by an activation layer with ReLU or ReLUx (llc) as activation function, a flatting layer to reshape (which is required for contrastive learning), followed by a linear layer (size: 512) and another activation layer. We include the flatting layer for all models since it is essential for contrastive learning, but did not had an effect on normally trained classifiers and we wanted to keep the architecture of the approaches as close as possible. The classifier consists of one linear layer. Implementation is done in Pytorch (Paszke et al., 2019). For contrastive learning (con, conN, conA and conF) we use stochastic gradient descent as optimizer as proposed by (Khosla et al., 2020), all other models are optimized using Adam optimizer. As mixing coefficient for CE-loss and feature-distance loss (feD training) we chose $\lambda = 0.1$ as proposed by (Chen et al., 2021). Hyper-parameters are determined by performing a grid-search. For training the contrastive models (con, conN, conA, Fcon) the grid search is done in [0.005; 0.1] and for the other models in [0.0001; 0.001]. All models are trained for 800 epochs and retrained for 200 epochs. The target retraining learning rate is search in [0.0001; 0.5] and we do are warm-start, i.e. we do not randomly internalize the weights of the classifier before target retraining. Model fine-tuning (conF) is done for 100 epochs and with a learning rate in [0.0001; 0.001].

**Datasets.** We use six different datasets: The SVHN dataset (Netzer et al., 2011) contains images of street few housing numbers. The MNIST dataset (LeCun & Cortes, 2010) consists of gray-scale images of handwritten digits. The CIFAR10 dataset (Krizhevsky et al., 2009) contains $3 \times 32 \times 32$ images of objects (airplane, bird, car, cat, deer, dog, horse, ship, truck and frog). The FashionMNIST/FMNIST dataset (Xiao et al., 2017) contains gray-scale images of clothes. The EMNIST dataset (Cohen et al., 2017) consists the 26 character of the alphabet and the KMNIST (Clanuwat et al., 2018) dataset consists of gray-scale images of Japanese characters. Each dataset already consists of a predefined training set and a test set. We further split the training sets into training data (90%) and validation data (10%).The validation set is used during training/retraining to check the accuracy and determine the best model. The following data augmentations are used on all datasets and model: random horizontal filps, random corps, and random rotations ($\leq 15°$). For contrastive models (con, conN, conA and conF) additional augmentations, i.e. random resize crop, color jitter and random gray-scale) is used.

**Transfer Learning Tasks** Based on the six datasets discussed above we create three transfer learning tasks of different relatedness. We consider the following transfer tasks (source domain $\rightarrow$ target domain): SVHN $\rightarrow$ MNIST (highly related), CIFAR10 $\rightarrow$ FMNIST (related), EMNIST $\rightarrow$ KMNIST (related).

**Perturbations and Attacks.** We use four different attack types: Noise attacks, Fast Gradient Sign Method (FGSM), Project Gradient Descent (PGD) attacks and DeepFool attacks with attack radii of 0.1. We chose an attack radius of 0.1 since that is a popular perturbation size analyzed in randomized smoothing as well as for attacks. The perturbation is bounded by the $L_2$-norm and applied to the input after data normalization. For adversarial training we use 10-step PGD attacks, while robustness analysis uses 1-step FGSM attacks, 100-step PGD attacks and 100-step DeepFool attacks based on the implementation provided by Rauber et al. (2020).

**Randomized Smoothing.** Randomized smoothing techniques (Cohen et al., 2019) draw samples $x_i \sim \mathcal{N}(x, \sigma)$ from the close neighborhood of input $x$, propagate them through the neural network and aggregate the outputs to obtain a smooth prediction. We use $\sigma = 0.1$, draw 500 samples for each input and bind the probability of returning an incorrect answer/prediction by $\alpha = 10^{-4}$. If a prediction cannot be certified randomized smoothing abstains, i.e. returns $-1$ instead of a label. Please note that the certified prediction mainly depends depends on the neighborhood of the input sample $x$ (i.e. the 500 drawn samples) and might be different than the prediction of the base model, which only depends on input $x$.

**Distribution shifts.** We generate different distribution shifts on each dataset based on random perturbations (noise), changes of the contrast (contrast) and Gaussian blur (blur). More specifically, we use Gaussian noise shift (Noise), uniform noise shift (UNoise), contrast reduction shift (Contrast), contrast reduction shift based on a binary search (ContrastBin), contrast reduction based on a linear

search(ContrastLin), Gaussin blur (Blur) and a salt and pepper (SaltPepper) shift. The perturbation or shift size is bounded by the $L_2$-norm and 5 as upper bound. To compute these perturbations we use the implementation provided by Rauber et al. (2020).

**Transferability.** In order to estimate transferability of source models we compute the H-score as proposed by (Bao et al., 2019).

## A.2 Additional experimental results

**Which training procedure results in the most robust source models?**

Table 1 shows how robust target model retrained on normal (ce), randomly perturbed (ceN) and adversarial perturbed (ceA) target inputs are. In order to quantify robustness we compute the verifiable accuracy using randomized smoothing and accuracy under FGSM, PGD and DeepFool attacks.

Table 1: Accuracy on the clean test set (base), verifiable accuracy (cert.) and radius, accuracy under FGSM, PGD and DeepFool attacks of source models trained on SVHN, EMNIST and CIFAR10. The best values/highest accuracy are highlighted using bold letters.

| | | | | SVHN $\rightarrow$ MNIST | | | |
|---|---|---|---|---|---|---|---|
| Train | Retrain | $A_{base}$ [%] | $A_{cert.}$ [%] | $R_{cert.}$ | $A_{FGSM}$ [%] | $A_{PGD}$ [%] | $A_{DeepFool}$ [%] |
| ce | – | 94.03 | 78.41 | 0.110 | 78.53 | 71.25 | 75.11 |
| ceN | – | 93.40 | **91.65** | 0.171 | 86.90 | 86.26 | 86.47 |
| ceA | – | 94.20 | 87.55 | 0.147 | 86.50 | 85.05 | 85.76 |
| feD | – | 93.69 | 88.29 | 0.157 | 86.70 | 86.17 | 86.38 |
| feA | – | 94.33 | 87.91 | 0.148 | 87.33 | 86.13 | 86.66 |
| llc | – | 77.60 | 76.00 | 0.165 | 72.66 | 72.71 | 71.50 |
| con | – | 93.32 | 69.91 | 0.096 | 79.35 | 66.73 | 75.06 |
| conN | – | 91.70 | 88.49 | 0.167 | 83.90 | 83.00 | 83.30 |
| conA | – | 92.41 | 85.27 | 0.145 | 83.75 | 82.16 | 82.96 |
| conF | – | **95.28** | 86.55 | 0.147 | **88.54** | **87.20** | **88.06** |
| | | | | EMNIST $\rightarrow$ KMNIST | | | |
| Train | Retrain | $A_{base}$ [%] | $A_{cert.}$ [%] | $R_{cert.}$ | $A_{FGSM}$ [%] | $A_{PGD}$ [%] | $A_{DeepFool}$ [%] |
| ce | – | 92.12 | 17.16 | 0.029 | 85.31 | 46.36 | 39.68 |
| ceN | – | 92.57 | **92.42** | 0.202 | 91.43 | 91.41 | 91.41 |
| ceA | – | 92.40 | 53.11 | 0.138 | 90.97 | 90.82 | 90.89 |
| feD | – | 92.29 | 89.63 | 0.198 | 91.40 | 91.39 | 91.40 |
| feA | – | 92.91 | 52.91 | 0.131 | 91.63 | 91.55 | 91.58 |
| llc | – | 91.06 | 90.96 | 0.204 | 90.35 | 90.30 | 90.27 |
| con | – | 92.63 | 8.81 | 0.094 | 81.23 | 26.64 | 56.16 |
| conN | – | 92.39 | 92.13 | 0.199 | 90.83 | 90.40 | 90.39 |
| conA | – | 92.34 | 11.88 | 0.130 | 90.74 | 90.49 | 90.65 |
| conF | – | **93.26** | 81.78 | 0.169 | **91.97** | **91.88** | **91.91** |
| | | | | CIFAR10 $\rightarrow$ FMNIST | | | |
| Train | Retrain | $A_{base}$ [%] | $A_{cert.}$ [%] | $R_{cert.}$ | $A_{FGSM}$ [%] | $A_{PGD}$ [%] | $A_{DeepFool}$ [%] |
| ce | – | 85.70 | 32.06 | 0.076 | 59.91 | 50.72 | 55.31 |
| ceN | – | 79.84 | 78.42 | 0.153 | 69.86 | 69.01 | 69.28 |
| ceA | – | 84.83 | 56.72 | 0.121 | 71.91 | 70.36 | 71.29 |
| feD | – | 83.46 | 64.58 | 0.127 | 72.66 | 71.49 | 72.14 |
| feA | – | 84.35 | 60.59 | 0.126 | 73.67 | 72.56 | 73.12 |
| llc | – | 65.75 | 62.44 | 0.172 | 61.34 | 61.41 | 60.79 |
| con | – | 87.26 | 31.43 | 0.083 | 61.14 | 51.84 | 57.41 |
| conN | – | 83.59 | **79.41** | 0.153 | 70.45 | 69.10 | 69.28 |
| conA | – | 85.39 | 51.29 | 0.114 | 72.72 | 71.44 | 71.47 |
| conF | – | **89.91** | 45.46 | 0.114 | **78.17** | **76.47** | **77.38** |

**Does robust retraining on the target domain improve robustness?**

Table 2, 3, 4 and Figure 8 show how target retraining affects robustness. In order to quantify robustness we compute the verifiable accuracy using randomized smoothing and accuracy under FGSM, PGD and DeepFool attacks.

Table 2: Accuracy on the clean test set (base), verifiable accuracy (cert.) and raduis, accuracy under FGSM, PGD and DeepFool attacks of target models trained on SVHN and retrained on MNIST. The best values/highest accuracy are highlighted using bold letters.

| Train | Retrain | SVHN $\rightarrow$ MNIST | | | | | |
|-------|---------|------------|------------|-----------|-------------|------------|------------------|
| | | $A_{base}$ [%] | $A_{cert.}$ [%] | $R_{cert.}$ | $A_{FGSM}$ [%] | $A_{PGD}$ [%] | $A_{DeepFool}$ [%] |
| ce | $R_{ce}$ | 71.07 | 8.09 | 0.002 | 23.97 | 1.34 | 1.07 |
| ce | $R_{ceN}$ | 40.29 | 31.47 | 0.017 | 9.70 | 0.27 | 0.14 |
| ce | $R_{ceA}$ | 61.26 | 9.90 | 0.009 | 34.49 | 4.43 | 2.33 |
| ceN | $R_{ce}$ | 70.31 | 69.12 | 0.121 | 57.67 | 53.13 | 53.08 |
| ceN | $R_{ceN}$ | 71.02 | 71.46 | 0.127 | 59.38 | 55.08 | 54.70 |
| ceN | $R_{ceA}$ | 69.72 | 68.60 | 0.121 | 57.33 | 52.75 | 52.79 |
| ceA | $R_{ce}$ | 60.40 | 36.28 | 0.054 | 45.88 | 22.55 | 34.80 |
| ceA | $R_{ceN}$ | 53.00 | 73.21 | 0.117 | 42.15 | 27.14 | 33.93 |
| ceA | $R_{ceA}$ | 60.42 | 33.96 | 0.052 | 46.22 | 24.48 | 35.90 |
| feD | $R_{ce}$ | 78.92 | 75.20 | 0.141 | 71.53 | 67.48 | 69.38 |
| feD | $R_{ceN}$ | 79.29 | 81.34 | 0.151 | 71.94 | 67.63 | 69.31 |
| feD | $R_{ceA}$ | 79.15 | 75.38 | 0.141 | 72.01 | 67.90 | 69.73 |
| feA | $R_{ce}$ | 59.64 | 50.95 | 0.088 | 46.20 | 38.65 | 38.30 |
| feA | $R_{ceN}$ | 53.10 | 66.12 | 0.098 | 40.13 | 33.69 | 32.96 |
| feA | $R_{ceA}$ | 59.09 | 51.09 | 0.091 | 46.82 | 40.46 | 39.44 |
| llc | $R_{ce}$ | 96.40 | 96.13 | 0.196 | 95.83 | 95.73 | 95.72 |
| llc | $R_{ceN}$ | 96.41 | **96.32** | 0.200 | **95.91** | **95.86** | **95.83** |
| llc | $R_{ceA}$ | **96.43** | 96.15 | 0.197 | 95.83 | 95.75 | 95.70 |
| con | $R_{ce}$ | 95.01 | 93.59 | 0.180 | 90.78 | 89.17 | 89.41 |
| con | $R_{ceN}$ | 94.58 | 93.90 | 0.184 | 89.93 | 88.32 | 88.71 |
| con | $R_{ceA}$ | 94.82 | 93.63 | 0.183 | 91.03 | 89.49 | 89.81 |
| conN | $R_{ce}$ | 94.58 | 94.12 | 0.193 | 92.33 | 91.85 | 91.91 |
| conN | $R_{ceN}$ | 94.45 | 93.86 | 0.193 | 92.09 | 91.55 | 91.59 |
| conN | $R_{ceA}$ | 94.65 | 94.05 | 0.192 | 92.38 | 91.85 | 91.83 |
| conA | $R_{ce}$ | 93.88 | 92.86 | 0.190 | 91.17 | 90.74 | 90.76 |
| conA | $R_{ceN}$ | 93.49 | 93.09 | 0.191 | 90.96 | 90.63 | 90.60 |
| conA | $R_{ceA}$ | 93.60 | 92.94 | 0.190 | 91.07 | 90.75 | 90.81 |
| conF | $R_{ce}$ | 85.08 | 84.86 | 0.145 | 75.75 | 71.95 | 71.96 |
| conF | $R_{ceN}$ | 83.05 | 84.81 | 0.147 | 72.32 | 68.16 | 68.59 |
| conF | $R_{ceA}$ | 86.14 | 86.04 | 0.144 | 76.10 | 71.94 | 72.18 |

Table 3: Accuracy on the clean test set (base), verifiable accuracy (cert.) and radius, accuracy under FGSM, PGD and DeepFool and attacks of target models trained on EMNIST and retrained on KMNIST. The best values/highest accuracy are highlighted using bold letters.

| Train | Retrain | $A_{base}$ [%] | $A_{cert.}$ [%] | $R_{cert.}$ | $A_{FGSM}$ [%] | $A_{PGD}$ [%] | $A_{DeepFool}$ [%] |
|-------|---------|-----------|-----------|-------|-----------|----------|--------------|
| ce | $R_{ce}$ | 79.29 | 36.82 | 0.030 | 63.82 | 19.79 | 15.80 |
| ce | $R_{ceN}$ | 41.00 | 57.63 | 0.059 | 27.91 | 6.98 | 4.70 |
| ce | $R_{ceA}$ | 78.08 | 38.97 | 0.039 | 65.77 | 29.14 | 23.06 |
| ceN | $R_{ce}$ | 81.96 | 81.57 | 0.188 | 79.36 | 79.21 | 79.14 |
| ceN | $R_{ceN}$ | 82.18 | 81.80 | 0.188 | 79.41 | 79.30 | 79.21 |
| ceN | $R_{ceA}$ | 82.19 | 81.71 | 0.188 | 79.45 | 79.31 | 79.26 |
| ceA | $R_{ce}$ | 80.71 | 44.83 | 0.123 | 76.30 | 75.86 | 75.96 |
| ceA | $R_{ceN}$ | 63.81 | 71.07 | 0.144 | 57.77 | 57.12 | 57.23 |
| ceA | $R_{ceA}$ | 80.65 | 44.73 | 0.124 | 76.40 | 75.87 | 76.04 |
| feD | $R_{ce}$ | **84.82** | 79.64 | 0.187 | 82.99 | 82.98 | 82.91 |
| feD | $R_{ceN}$ | 82.04 | **83.46** | 0.190 | 79.91 | 79.89 | 79.79 |
| feD | $R_{ceA}$ | 84.76 | 79.86 | 0.187 | **83.05** | **83.04** | **82.97** |
| feA | $R_{ce}$ | 82.09 | 41.82 | 0.142 | 78.34 | 78.09 | 78.07 |
| feA | $R_{ceN}$ | 66.10 | 75.17 | 0.158 | 60.85 | 60.46 | 60.35 |
| feA | $R_{ceA}$ | 82.25 | 44.98 | 0.137 | 78.45 | 78.17 | 78.13 |
| llc | $R_{ce}$ | 81.21 | 81.15 | 0.191 | 79.61 | 79.41 | 79.30 |
| llc | $R_{ceN}$ | 80.96 | 80.83 | 0.191 | 79.03 | 78.85 | 78.76 |
| llc | $R_{ceA}$ | 81.26 | 81.19 | 0.191 | 79.72 | 79.50 | 79.36 |
| con | $R_{ce}$ | 69.69 | 13.67 | 0.066 | 46.34 | 7.24 | 8.65 |
| con | $R_{ceN}$ | 18.45 | 39.21 | 0.050 | 12.47 | 2.12 | 2.50 |
| con | $R_{ceA}$ | 62.56 | 17.70 | 0.039 | 47.16 | 11.48 | 12.80 |
| conN | $R_{ce}$ | 60.88 | 58.67 | 0.126 | 50.73 | 47.39 | 47.25 |
| conN | $R_{ceN}$ | 60.39 | 58.64 | 0.128 | 49.90 | 47.26 | 47.13 |
| conN | $R_{ceA}$ | 61.16 | 58.98 | 0.129 | 51.30 | 48.35 | 48.13 |
| conA | $R_{ce}$ | 61.80 | 12.26 | 0.168 | 52.30 | 49.85 | 50.17 |
| conA | $R_{ceN}$ | 20.62 | 38.35 | 0.092 | 14.25 | 13.08 | 13.59 |
| conA | $R_{ceA}$ | 60.93 | 11.41 | 0.178 | 51.28 | 49.32 | 49.45 |
| conF | $R_{ce}$ | 51.86 | 35.58 | 0.089 | 43.35 | 42.03 | 41.84 |
| conF | $R_{ceN}$ | 41.81 | 40.54 | 0.089 | 33.19 | 31.92 | 31.86 |
| conF | $R_{ceA}$ | 51.81 | 35.84 | 0.094 | 43.75 | 42.46 | 42.22 |

Table 4: Accuracy on the clean test set (base), verifiable accuracy (cert.) and radius, accuracy under FGSM, PGD and DeepFool attacks of target models trained on CIFAR10 and retrained on FMNIST. The best values/highest accuracy are highlighted using bold letters.

| Train | Retrain | $A_{base}$ [%] | $A_{cert.}$ [%] | $R_{cert.}$ | $A_{FGSM}$ [%] | $A_{PGD}$ [%] | $A_{DeepFool}$ [%] |
|-------|---------|------|------|------|------|------|------|
| ce | $R_{ce}$ | 71.11 | 26.25 | 0.039 | 47.72 | 28.13 | 31.91 |
| ce | $R_{ceN}$ | 35.64 | 64.96 | 0.084 | 16.98 | 6.92 | 8.37 |
| ce | $R_{ceA}$ | 70.47 | 26.03 | 0.036 | 51.57 | 36.20 | 38.34 |
| ceN | $R_{ce}$ | 73.18 | 69.58 | 0.144 | 64.32 | 63.11 | 63.17 |
| ceN | $R_{ceN}$ | 72.48 | 71.33 | 0.150 | 64.49 | 63.31 | 63.53 |
| ceN | $R_{ceA}$ | 73.23 | 69.44 | 0.146 | 64.67 | 63.58 | 63.63 |
| ceA | $R_{ce}$ | 74.60 | 67.61 | 0.131 | 63.63 | 61.81 | 62.15 |
| ceA | $R_{ceN}$ | 65.87 | 72.10 | 0.155 | 56.58 | 54.81 | 55.13 |
| ceA | $R_{ceA}$ | 75.32 | 66.70 | 0.132 | 65.37 | 63.71 | 64.17 |
| feD | $R_{ce}$ | 71.52 | 57.24 | 0.118 | 62.19 | 60.41 | 60.96 |
| feD | $R_{ceN}$ | 59.31 | 70.05 | 0.159 | 50.89 | 49.25 | 49.66 |
| feD | $R_{ceA}$ | 71.92 | 58.02 | 0.119 | 63.30 | 61.71 | 62.26 |
| feA | $R_{ce}$ | 69.83 | 61.53 | 0.108 | 59.44 | 57.08 | 57.56 |
| feA | $R_{ceN}$ | 64.08 | 70.39 | 0.137 | 54.99 | 52.89 | 53.28 |
| feA | $R_{ceA}$ | 69.82 | 61.12 | 0.113 | 60.29 | 58.38 | 58.69 |
| llc | $R_{ce}$ | 80.07 | 79.70 | 0.180 | 76.08 | 75.80 | 75.64 |
| llc | $R_{ceN}$ | 79.02 | 79.28 | 0.184 | 75.93 | 75.70 | 75.59 |
| llc | $R_{ceA}$ | 80.10 | **79.88** | 0.185 | **76.64** | **76.45** | **76.38** |
| con | $R_{ce}$ | 79.81 | 38.93 | 0.076 | 51.31 | 28.42 | 33.61 |
| con | $R_{ceN}$ | 70.39 | 73.22 | 0.122 | 32.51 | 14.57 | 16.96 |
| con | $R_{ceA}$ | 78.13 | 48.20 | 0.092 | 57.73 | 41.66 | 44.26 |
| conN | $R_{ce}$ | **82.25** | 78.26 | 0.158 | 68.31 | 63.39 | 63.98 |
| conN | $R_{ceN}$ | 80.94 | 78.42 | 0.163 | 67.43 | 62.74 | 63.44 |
| conN | $R_{ceA}$ | 82.04 | 78.35 | 0.165 | 70.27 | 66.87 | 67.15 |
| conA | $R_{ce}$ | 80.31 | 53.23 | 0.098 | 67.66 | 64.09 | 65.01 |
| conA | $R_{ceN}$ | 73.11 | 73.07 | 0.132 | 61.69 | 58.02 | 58.99 |
| conA | $R_{ceA}$ | 79.75 | 54.87 | 0.102 | 68.31 | 65.41 | 66.06 |
| conF | $R_{ce}$ | 75.97 | 61.55 | 0.097 | 60.67 | 56.97 | 58.09 |
| conF | $R_{ceN}$ | 62.25 | 70.72 | 0.120 | 42.03 | 38.47 | 39.68 |
| conF | $R_{ceA}$ | 75.36 | 64.26 | 0.105 | 61.96 | 59.15 | 59.79 |

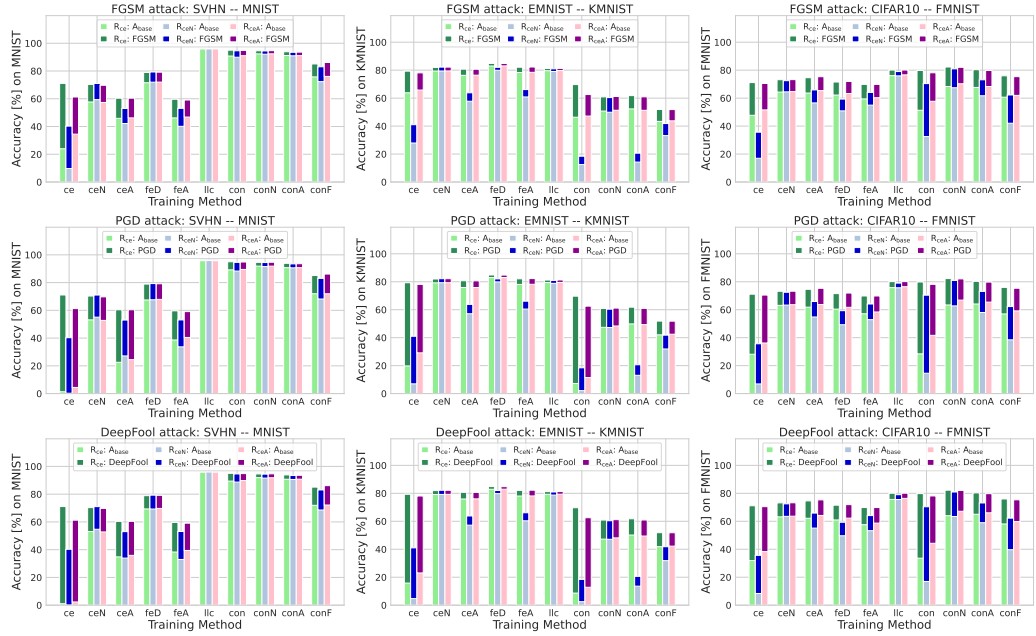

Figure 8: Accuracy on the clean test set (base, bright colors), verifiable accuracy (cert., dark colors), accuracy under FGSM (dark green), DeepFool (DF, dark blue) and PGD attacks (dark purple) of target models retrained on normal ($R_{ce}$), randomly ($R_{ceN}$) or adversarially perturbed ($R_{ceA}$) target data.

**Which training/target retraining provides models that are robust against distribution shifts?**

Table 5, 6, 7, 8 and Figure 9, 10, 11, 12 show accuracy of source models and target models under distribution shifts based on noise perturbations, changes in contrast and Gaussian blur. More specifically, we use Gaussian noise shift (Noise), uniform noise shift (UNoise), contrast reduction shift (Contrast), contrast reduction shift based on a binary search (ContrastBin), contrast reduction based on a linear search(ContrastLin), Gaussin blur (Blur) and a salt and pepper (SaltPepper) shift.

Table 5: Accuracy on the clean test set ($A_{base}$) and under distribution shifts of source models trained on SVHN (first row), EMNIST (second row) and CIFAR10 (third row). The best values/highest accuracy are highlighted using bold letters.

| | | | | | | SVHN → MNIST | | | |
|---|---|---|---|---|---|---|---|---|---|
| Train | Retrain | $A_{base}$ [%] | $A_{Noise}$ [%] | $A_{UNoise}$ [%] | $A_{Contrast}$ [%] | $A_{ContrastBin}$ [%] | $A_{ContrastLin}$ [%] | $A_{Blur}$ [%] | $A_{SaltPepper}$ [%] |
| ce | – | 94.03 | **82.45** | 81.91 | 86.37 | 84.17 | 83.90 | 25.99 | 29.70 |
| ceN | – | 93.40 | **91.97** | **91.66** | 80.83 | 79.11 | 78.68 | 26.67 | 36.62 |
| ceA | – | 94.20 | 88.97 | 88.78 | 85.87 | 84.55 | 84.33 | 29.29 | 29.23 |
| feD | – | 93.69 | 89.59 | 89.57 | 85.16 | 83.67 | 83.49 | 28.54 | 38.10 |
| feA | – | 94.33 | 89.39 | 89.33 | 85.66 | 83.99 | 83.80 | 28.08 | 32.85 |
| llc | – | 77.60 | 76.76 | 76.76 | 61.54 | 60.46 | 60.46 | 29.16 | **48.19** |
| con | – | 93.32 | 77.04 | 76.65 | 87.91 | **87.06** | **86.97** | 25.99 | 21.87 |
| conN | – | 91.70 | 88.95 | 88.82 | 85.21 | 84.10 | 84.04 | 26.30 | 38.67 |
| conA | – | 92.41 | 87.32 | 87.05 | 86.07 | 85.02 | 84.96 | 26.86 | 25.15 |
| conF | – | **95.28** | 88.66 | 88.63 | **88.15** | 86.81 | 86.70 | **29.51** | 38.49 |

| | | | | | | EMNIST → KMNIST | | | |
|---|---|---|---|---|---|---|---|---|---|
| Train | Retrain | $A_{base}$ [%] | $A_{Noise}$ [%] | $A_{UNoise}$ [%] | $A_{Contrast}$ [%] | $A_{ContrastBin}$ [%] | $A_{ContrastLin}$ [%] | $A_{Blur}$ [%] | $A_{SaltPepper}$ [%] |
| ce | – | 92.12 | 25.29 | 22.93 | 11.34 | 9.33 | 8.38 | 27.80 | 42.10 |
| ceN | – | 92.57 | 92.53 | 92.60 | 82.51 | 80.72 | 80.27 | 87.05 | 75.27 |
| ceA | – | 92.40 | 83.20 | 83.25 | 20.38 | 19.60 | 18.69 | 71.82 | 29.46 |
| feD | – | 92.29 | 91.72 | 91.79 | 89.74 | 88.32 | 88.17 | **88.51** | 61.76 |
| feA | – | 92.91 | **92.82** | **92.82** | 92.45 | 91.71 | 91.65 | 83.14 | 41.47 |
| llc | – | 91.06 | 91.07 | 91.00 | 90.27 | 89.29 | 89.27 | 85.39 | **86.87** |
| con | – | 92.63 | 15.90 | 16.54 | 92.54 | 92.08 | 91.84 | 83.53 | 4.59 |
| conN | – | 92.39 | 92.29 | 92.31 | 92.36 | 91.95 | 91.87 | 85.63 | 18.13 |
| conA | – | 92.34 | 20.94 | 21.81 | 92.44 | 91.99 | 91.94 | 84.86 | 9.82 |
| conF | – | **93.26** | 92.73 | 92.79 | **93.30** | **92.74** | **92.71** | 86.34 | 41.03 |

| | | | | | | CIFAR10 → FMNIST | | | |
|---|---|---|---|---|---|---|---|---|---|
| Train | Retrain | $A_{base}$ [%] | $A_{Noise}$ [%] | $A_{UNoise}$ [%] | $A_{Contrast}$ [%] | $A_{ContrastBin}$ [%] | $A_{ContrastLin}$ [%] | $A_{Blur}$ [%] | $A_{SaltPepper}$ [%] |
| ce | – | 85.70 | 44.11 | 43.27 | 75.41 | 72.25 | 71.72 | 16.61 | 26.34 |
| ceN | – | 79.84 | 80.24 | 80.08 | 60.61 | 56.36 | 55.59 | 25.37 | 33.40 |
| ceA | – | 84.83 | 64.70 | 64.05 | 74.72 | 71.04 | 70.54 | 20.89 | 31.79 |
| feD | – | 83.46 | 71.00 | 70.54 | 69.95 | 65.85 | 65.24 | 24.96 | 29.78 |
| feA | – | 84.35 | 68.09 | 67.93 | 71.65 | 67.87 | 67.39 | 25.47 | 34.55 |
| llc | – | 65.75 | 63.77 | 63.70 | 52.64 | 46.95 | 46.88 | 35.97 | **50.25** |
| con | – | 87.26 | 42.46 | 41.33 | **86.43** | **85.08** | **85.00** | 43.09 | 38.48 |
| conN | – | 83.59 | **80.70** | **80.70** | 82.23 | 80.22 | 80.12 | **45.54** | 41.80 |
| conA | – | 85.39 | 59.45 | 58.54 | 84.36 | 82.67 | 82.63 | 44.96 | 41.47 |
| conF | – | **89.91** | 56.31 | 55.41 | 85.36 | 83.54 | 83.51 | 28.81 | 44.57 |

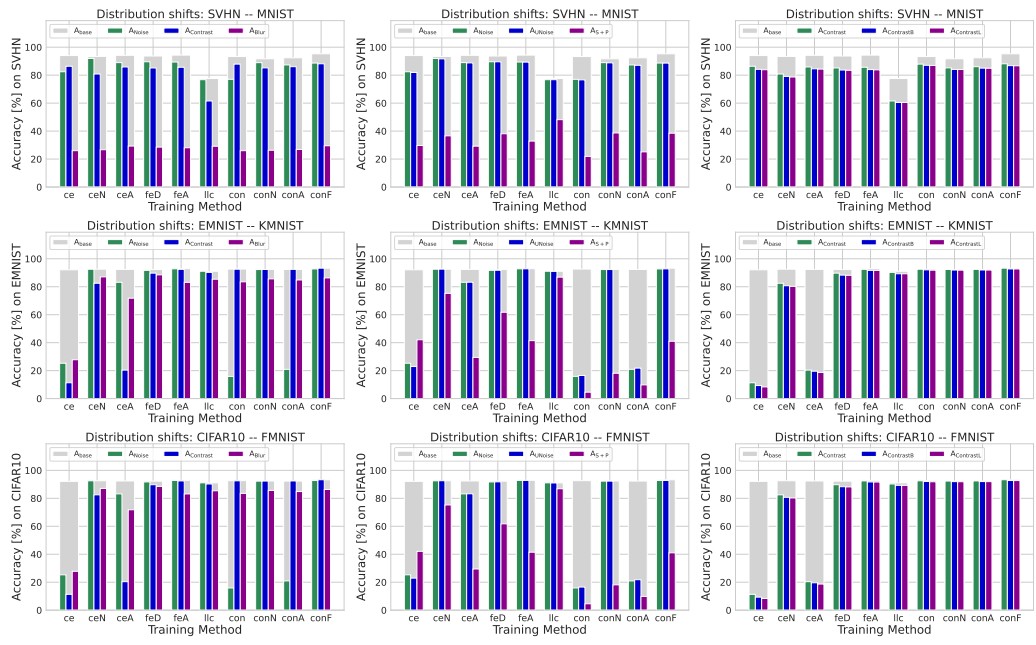

Figure 9: Accuracy on the clean test set ($A_{base}$, bright colors) and accuracy under distribution shifts based on random noise, changes of the contrast and Gaussian blur of target model trained on SVHN, EMNIST or CIFAR10.

Table 6: Accuracy on the clean test set ($A_{base}$) and under distribution shifts of the target models trained on SVHN and retrained on MNIST. The best values/highest accuracy are highlighted using bold letters.

| | | SVHN → MNIST | | | | | | | |
|---|---|---|---|---|---|---|---|---|---|
| Train | Retrain | $A_{base}$ [%] | $A_{Noise}$ [%] | $A_{UNoise}$ [%] | $A_{Contrast}$ [%] | $A_{ContrastBin}$ [%] | $A_{ContrastLin}$ [%] | $A_{Blur}$ [%] | $A_{SaltPepper}$ [%] |
| ce | $R_{ce}$ | 71.07 | 36.04 | 36.20 | 33.93 | 27.45 | 15.81 | 25.85 | 8.91 |
| ce | $R_{ceN}$ | 40.29 | 28.90 | 28.84 | 26.03 | 17.48 | 7.51 | 8.26 | 3.50 |
| ce | $R_{ceA}$ | 61.26 | 62.01 | 62.71 | 64.32 | 46.20 | 27.11 | 29.88 | 16.19 |
| ceN | $R_{ce}$ | 70.31 | 61.34 | 60.72 | 57.16 | 46.62 | 38.70 | 40.16 | 23.31 |
| ceN | $R_{ceN}$ | 71.02 | 67.94 | 67.80 | 64.99 | 51.71 | 45.00 | 44.04 | 26.72 |
| ceN | $R_{ceA}$ | 69.72 | 54.59 | 52.38 | 45.68 | 38.20 | 31.42 | 37.14 | 22.50 |
| ceA | $R_{ce}$ | 60.40 | 41.80 | 41.63 | 45.77 | 30.99 | 15.03 | 21.52 | 1.68 |
| ceA | $R_{ceN}$ | 53.00 | 81.85 | 82.85 | 81.83 | 48.00 | 36.70 | 32.82 | 8.42 |
| ceA | $R_{ceA}$ | 60.42 | 38.88 | 39.12 | 40.04 | 25.85 | 14.35 | 20.25 | 1.50 |
| feD | $R_{ce}$ | 78.92 | 43.52 | 41.10 | 40.17 | 32.67 | 17.11 | 44.46 | 17.06 |
| feD | $R_{ceN}$ | 79.29 | 55.34 | 52.87 | 46.16 | 38.54 | 23.91 | 51.90 | 16.88 |
| feD | $R_{ceA}$ | 79.15 | 42.94 | 40.66 | 39.12 | 31.80 | 17.03 | 44.12 | 17.98 |
| feA | $R_{ce}$ | 59.64 | 51.24 | 50.50 | 50.36 | 38.61 | 28.96 | 29.64 | 5.10 |
| feA | $R_{ceN}$ | 53.10 | 56.20 | 54.80 | 56.84 | 36.33 | 28.56 | 32.61 | 5.41 |
| feA | $R_{ceA}$ | 59.09 | 50.68 | 48.83 | 46.96 | 35.12 | 25.47 | 29.44 | 6.25 |
| llc | $R_{ce}$ | 96.40 | **96.57** | **96.56** | 96.60 | 95.97 | 95.96 | 91.79 | 92.54 |
| llc | $R_{ceN}$ | **96.41** | 96.47 | 96.43 | 96.53 | **96.10** | **96.09** | 93.37 | **93.12** |
| llc | $R_{ceA}$ | 96.43 | 96.52 | 96.47 | **96.68** | 96.06 | 96.05 | 92.03 | 92.58 |
| con | $R_{ce}$ | 95.01 | 94.17 | 94.47 | 95.34 | 94.39 | 94.29 | 91.21 | 24.79 |
| con | $R_{ceN}$ | 94.58 | 94.37 | 94.38 | 94.73 | 93.88 | 93.80 | 90.59 | 29.13 |
| con | $R_{ceA}$ | 94.82 | 94.17 | 94.13 | 95.05 | 94.12 | 94.08 | 91.27 | 29.82 |
| conN | $R_{ce}$ | 94.58 | 94.74 | 94.78 | 95.00 | 93.68 | 93.60 | 90.45 | 47.45 |
| conN | $R_{ceN}$ | 94.45 | 94.48 | 94.47 | 94.65 | 93.45 | 93.35 | 90.27 | 50.45 |
| conN | $R_{ceA}$ | 94.65 | 94.69 | 94.79 | 94.96 | 93.72 | 93.63 | 90.43 | 47.82 |
| conA | $R_{ce}$ | 93.88 | 93.82 | 93.82 | 94.08 | 93.28 | 93.19 | 89.68 | 34.87 |
| conA | $R_{ceN}$ | 93.49 | 93.69 | 93.53 | 93.89 | 92.94 | 92.89 | 89.33 | 35.91 |
| conA | $R_{ceA}$ | 93.60 | 93.35 | 93.62 | 94.05 | 93.06 | 93.02 | 89.35 | 32.86 |
| conF | $R_{ce}$ | 85.08 | 37.97 | 37.13 | 34.56 | 32.12 | 23.09 | 42.45 | 30.06 |
| conF | $R_{ceN}$ | 83.05 | 34.96 | 33.39 | 29.20 | 26.60 | 16.57 | 34.18 | 31.33 |
| conF | $R_{ceA}$ | 86.14 | 31.82 | 30.66 | 28.35 | 26.02 | 16.42 | 34.99 | 27.15 |

Table 7: Accuracy on the clean test set ($A_{base}$) and under distribution shifts of the target models trained on EMNIST and retrained on KMNIST. The best values/highest accuracy are highlighted using bold letters.

| | | EMNIST → KMNIST | | | | | | | |
|---|---|---|---|---|---|---|---|---|---|
| Train | Retrain | $A_{base}$ [%] | $A_{Noise}$ [%] | $A_{UNoise}$ [%] | $A_{Contrast}$ [%] | $A_{ContrastBin}$ [%] | $A_{ContrastLin}$ [%] | $A_{Blur}$ [%] | $A_{SaltPepper}$ [%] |
| ce | $R_{ce}$ | 79.29 | 31.75 | 30.54 | 19.19 | 17.68 | 16.14 | 36.07 | 37.09 |
| ce | $R_{ceN}$ | 41.00 | 31.43 | 30.15 | 18.69 | 13.44 | 12.44 | 19.18 | 23.87 |
| ce | $R_{ceA}$ | 78.08 | 34.14 | 32.82 | 19.61 | 18.27 | 16.64 | 40.84 | 43.26 |
| ceN | $R_{ce}$ | 81.96 | 82.09 | 81.90 | 71.06 | 67.67 | 66.98 | 71.08 | 61.18 |
| ceN | $R_{ceN}$ | 82.18 | 82.06 | 81.98 | 71.52 | 68.12 | 67.32 | 71.07 | 61.42 |
| ceN | $R_{ceA}$ | 82.19 | 82.06 | 81.92 | 71.27 | 67.98 | 67.26 | 71.38 | 61.47 |
| ceA | $R_{ce}$ | 80.71 | 69.89 | 69.58 | 28.14 | 24.92 | 23.35 | 61.33 | 24.63 |
| ceA | $R_{ceN}$ | 63.81 | 70.98 | 71.42 | 61.43 | 47.05 | 43.16 | 44.55 | 20.37 |
| ceA | $R_{ceA}$ | 80.65 | 69.51 | 69.68 | 27.90 | 24.64 | 23.22 | 61.33 | 25.23 |
| feD | $R_{ce}$ | **84.82** | 83.01 | **82.99** | 79.86 | 77.24 | 76.98 | **79.13** | 53.70 |
| feD | $R_{ceN}$ | 82.04 | **83.32** | **83.39** | **83.59** | 78.85 | 78.66 | 75.52 | 56.31 |
| feD | $R_{ceA}$ | 84.76 | 82.96 | 83.12 | 79.86 | 77.26 | 77.05 | 79.11 | 53.78 |
| feA | $R_{ce}$ | 82.09 | 81.31 | 81.66 | 80.69 | 78.69 | 78.44 | 70.02 | 35.69 |
| feA | $R_{ceN}$ | 66.10 | 67.35 | 66.45 | 66.08 | 62.50 | 62.09 | 50.61 | 29.25 |
| feA | $R_{ceA}$ | 82.25 | 81.46 | 81.35 | 80.80 | **78.92** | **78.71** | 70.56 | 36.85 |
| llc | $R_{ce}$ | 81.21 | 81.16 | 81.08 | 79.91 | 76.93 | 76.87 | 73.01 | 71.50 |
| llc | $R_{ceN}$ | 80.96 | 80.90 | 80.84 | 79.66 | 76.73 | 76.63 | 72.11 | 71.23 |
| llc | $R_{ceA}$ | 81.26 | 81.40 | 81.17 | 80.01 | 77.03 | 77.03 | 73.29 | **71.76** |
| con | $R_{ce}$ | 69.69 | 23.74 | 24.72 | 69.56 | 66.23 | 65.04 | 50.54 | 1.99 |
| con | $R_{ceN}$ | 18.45 | 43.19 | 42.43 | 17.48 | 16.03 | 15.51 | 10.97 | 2.75 |
| con | $R_{ceA}$ | 62.56 | 27.84 | 28.53 | 62.13 | 59.37 | 58.38 | 47.81 | 3.38 |
| conN | $R_{ce}$ | 60.88 | 60.75 | 60.99 | 60.85 | 57.15 | 56.41 | 42.91 | 2.26 |
| conN | $R_{ceN}$ | 60.39 | 60.50 | 60.38 | 60.28 | 56.65 | 55.94 | 42.73 | 2.45 |
| conN | $R_{ceA}$ | 61.16 | 61.01 | 61.23 | 60.72 | 57.30 | 56.70 | 43.17 | 2.76 |
| conA | $R_{ce}$ | 61.80 | 23.08 | 24.60 | 62.60 | 59.05 | 58.67 | 41.60 | 1.90 |
| conA | $R_{ceN}$ | 20.62 | 35.41 | 35.66 | 20.15 | 18.02 | 17.79 | 11.14 | 0.66 |
| conA | $R_{ceA}$ | 60.93 | 22.06 | 22.83 | 61.43 | 58.22 | 57.88 | 42.05 | 2.18 |
| conF | $R_{ce}$ | 51.86 | 46.70 | 47.97 | 50.85 | 46.81 | 46.40 | 33.94 | 3.36 |
| conF | $R_{ceN}$ | 41.81 | 44.11 | 44.01 | 43.28 | 38.02 | 37.48 | 24.82 | 3.04 |
| conF | $R_{ceA}$ | 51.81 | 47.63 | 48.10 | 51.42 | 47.28 | 46.86 | 34.90 | 4.45 |

Table 8: Accuracy on the clean test set ($A_{base}$) and under distribution shifts of the target models trained on CIFAR10 and retrained on FMNIST. The best values/highest accuracy are highlighted using bold letters.

| | | CIFAR10 → FMNIST | | | | | | | |
|---|---|---|---|---|---|---|---|---|---|
| Train | Retrain | $A_{base}$ [%] | $A_{Noise}$ [%] | $A_{UNoise}$ [%] | $A_{Contrast}$ [%] | $A_{ContrastBin}$ [%] | $A_{ContrastLin}$ [%] | $A_{Blur}$ [%] | $A_{SaltPepper}$ [%] |
| ce | $R_{ce}$ | 71.11 | 37.35 | 37.26 | 65.04 | 58.13 | 49.88 | 22.57 | 17.40 |
| ce | $R_{ceN}$ | 35.64 | 48.57 | 48.03 | 38.90 | 27.91 | 22.82 | 9.57 | 7.21 |
| ce | $R_{ceA}$ | 70.47 | 35.17 | 35.85 | 65.69 | 60.36 | 54.02 | 24.38 | 21.64 |
| ceN | $R_{ce}$ | 73.18 | 61.93 | 60.56 | 41.54 | 37.68 | 35.90 | 33.92 | 21.92 |
| ceN | $R_{ceN}$ | 72.48 | 66.49 | 65.33 | 44.60 | 40.87 | 38.48 | 35.95 | 24.23 |
| ceN | $R_{ceA}$ | 73.23 | 62.67 | 60.70 | 41.62 | 38.07 | 36.04 | 34.45 | 21.99 |
| ceA | $R_{ce}$ | 74.60 | 70.19 | 70.53 | 69.57 | 64.56 | 62.64 | 41.42 | 21.42 |
| ceA | $R_{ceN}$ | 65.87 | 72.63 | 72.65 | 68.57 | 61.20 | 58.75 | 37.25 | 28.63 |
| ceA | $R_{ceA}$ | 75.32 | 69.50 | 69.36 | 70.56 | 65.48 | 63.08 | 42.43 | 19.94 |
| feD | $R_{ce}$ | 71.52 | 60.90 | 60.68 | 56.94 | 53.64 | 49.80 | 36.16 | 13.69 |
| feD | $R_{ceN}$ | 59.31 | 69.01 | 69.03 | 61.87 | 52.33 | 50.82 | 36.03 | 16.16 |
| feD | $R_{ceA}$ | 71.92 | 61.23 | 61.63 | 56.76 | 53.46 | 50.00 | 37.21 | 15.03 |
| feA | $R_{ce}$ | 69.83 | 47.54 | 47.50 | 49.23 | 45.50 | 41.69 | 35.38 | 14.85 |
| feA | $R_{ceN}$ | 64.08 | 59.07 | 59.14 | 52.91 | 47.39 | 43.52 | 30.94 | 20.98 |
| feA | $R_{ceA}$ | 69.82 | 49.89 | 49.31 | 51.29 | 47.48 | 44.83 | 35.64 | 16.14 |
| llc | $R_{ce}$ | 80.07 | 79.39 | 79.13 | 71.15 | 69.52 | 69.38 | 65.79 | 67.24 |
| llc | $R_{ceN}$ | 79.02 | 78.55 | 78.52 | 69.56 | 67.88 | 67.81 | 65.07 | **68.27** |
| llc | $R_{ceA}$ | 80.10 | 79.33 | 79.05 | 70.92 | 69.36 | 69.23 | **66.18** | 67.88 |
| con | $R_{ce}$ | 79.81 | 48.53 | 49.73 | 79.71 | 76.19 | 75.86 | 52.91 | 28.03 |
| con | $R_{ceN}$ | 70.39 | 76.02 | 75.90 | 70.25 | 64.80 | 64.18 | 40.55 | 17.18 |
| con | $R_{ceA}$ | 78.13 | 57.63 | 58.49 | 77.29 | 74.91 | 74.77 | 58.66 | 36.11 |
| conN | $R_{ce}$ | **82.25** | 79.66 | 79.70 | **80.85** | 76.91 | 76.27 | 58.27 | 24.65 |
| conN | $R_{ceN}$ | 80.94 | **80.07** | **80.00** | 79.91 | 75.96 | 75.30 | 61.20 | 23.92 |
| conN | $R_{ceA}$ | 82.04 | 79.61 | 79.27 | 80.73 | **77.68** | **77.21** | 60.58 | 30.34 |
| conA | $R_{ce}$ | 80.31 | 64.51 | 64.98 | 77.09 | 74.48 | 74.23 | 59.60 | 11.86 |
| conA | $R_{ceN}$ | 73.11 | 75.74 | 75.85 | 71.64 | 69.01 | 68.63 | 51.17 | 14.07 |
| conA | $R_{ceA}$ | 79.75 | 64.68 | 65.91 | 77.26 | 74.80 | 74.61 | 60.82 | 12.35 |
| conF | $R_{ce}$ | 75.97 | 62.19 | 61.79 | 68.66 | 65.13 | 62.87 | 41.23 | 27.31 |
| conF | $R_{ceN}$ | 62.25 | 67.67 | 67.81 | 56.35 | 51.13 | 47.13 | 24.86 | 14.25 |
| conF | $R_{ceA}$ | 75.36 | 66.40 | 65.50 | 69.19 | 66.01 | 64.15 | 43.30 | 32.16 |

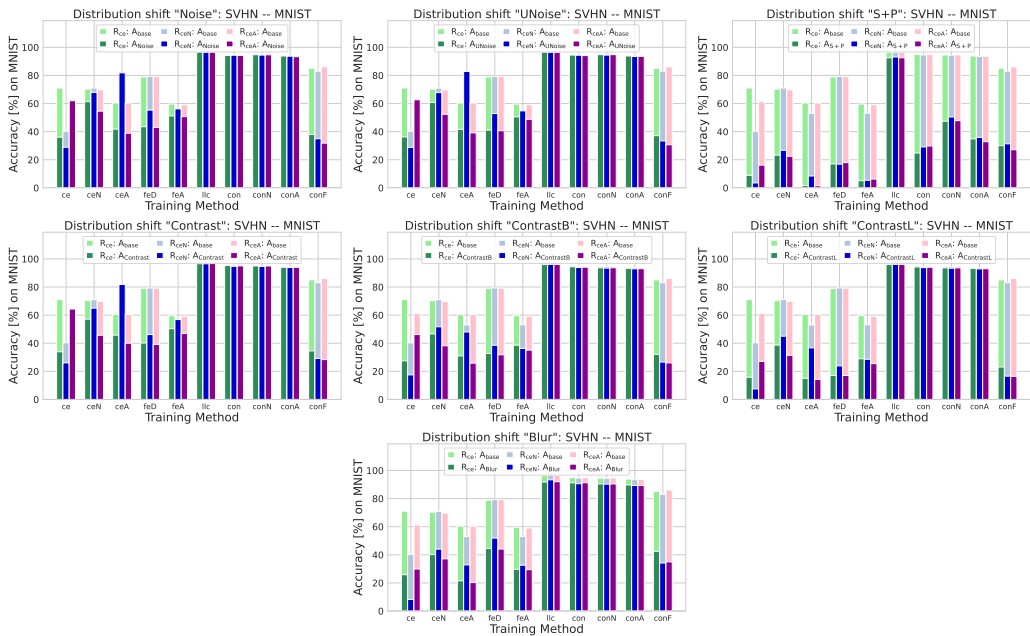

Figure 10: Accuracy on the clean test set ($A_{base}$, bright colors) and accuracy under distribution shifts based on random noise, changes of the contrast and Gaussian blur of target model trained on SVHN and retrained on clean ($R_{ce}$), randomly ($R_{ceN}$) or adversarial perturbed ($R_{ceA}$) inputs (MNIST).

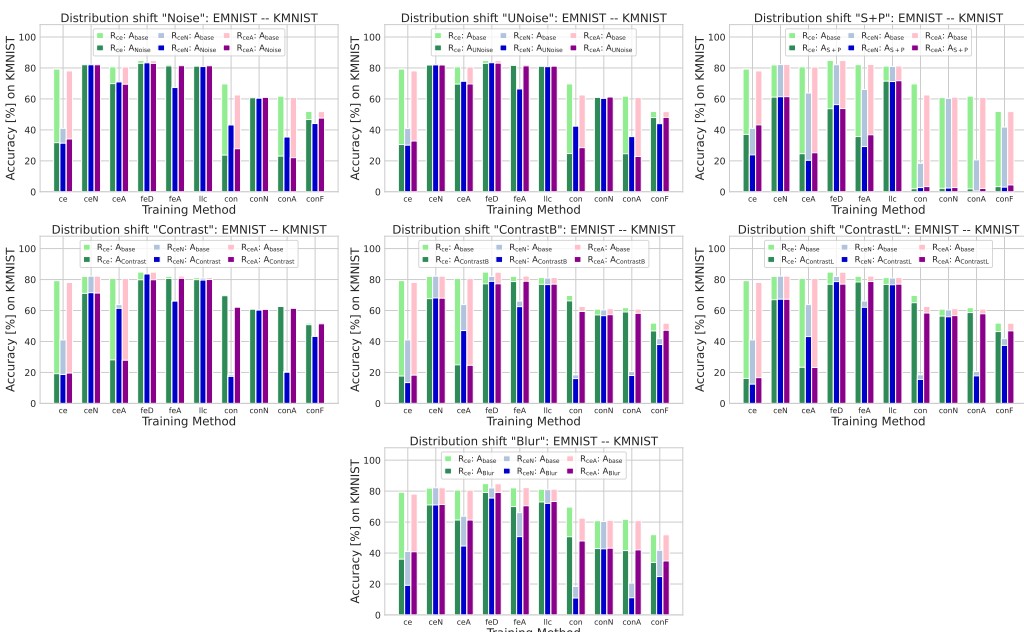

Figure 11: Accuracy on the clean test set ($A_{base}$, bright colors) and accuracy under distribution shifts based on random noise, changes of the contrast and Gaussian blur of target model trained on EMNIST and retrained on clean ($R_{ce}$), randomly ($R_{ceN}$) or adversarial perturbed ($R_{ceA}$) inputs (KMNIST).

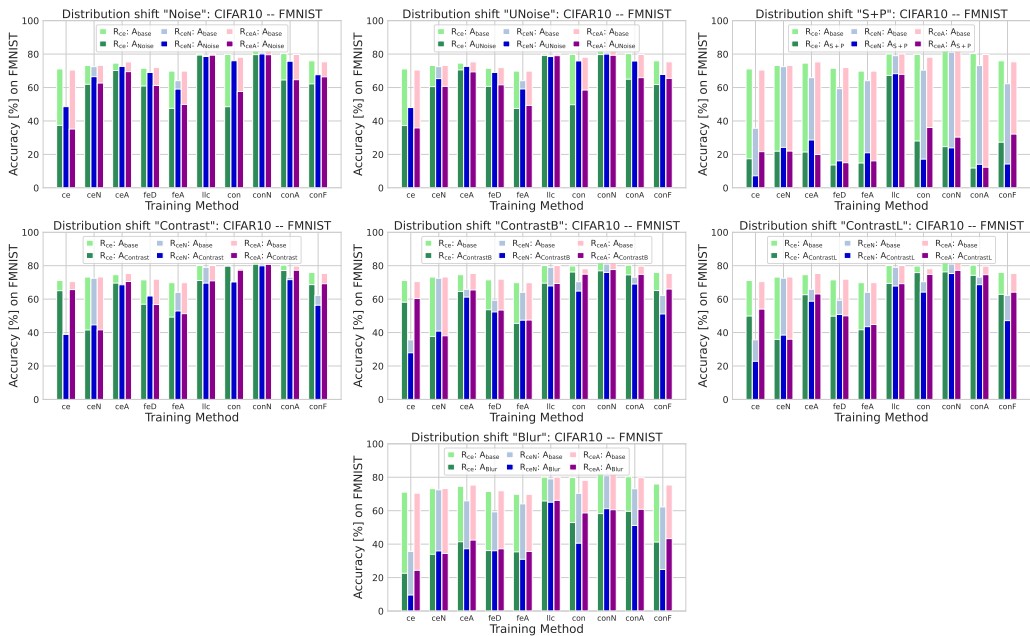

Figure 12: Accuracy on the clean test set ($A_{base}$, bright colors) and accuracy under distribution shifts based on random noise, changes of the contrast and Gaussian blur of target model trained on CIFAR10 and retrained on clean ($R_{ce}$), randomly ($R_{ceN}$) or adversarial perturbed ($R_{ceN}$) inputs (FMNIST).

**Does transferability correlate with model robustness?**

Table 9 shows the verifiable accuracy ($A_{Smooth}$), accuracy under the strongest (FGSM, PGD, Deep-Fool) attack ($A_{str.\ attack}$) and quantifies transferability using H-score. Table 10 shows the source accuracy and the zero-shot target accuracy, i.e. the accuracy a source model achieves on the target dataset before target retraining. Results of both tables are visualized in Figure 13.

Table 9: Transferability, verifiable accuracy and accuracy under attack of source models versus robustness on the source dataset. Transferability is measured by the H-Score, while robustness is quantified using verifiable accuracy and accuracy under the strongest attack.

| | SVHN → MNIST | | | EMNIST → KMNIST | | | CIFAR10 → FMNIST | | |
|---|---|---|---|---|---|---|---|---|---|
| Train | $A_{cert.}$ [%] | $A_{str.\ attack}$ [%] | $H-score$ | $A_{cert.}$ [%] | $A_{str.\ attack}$ [%] | $H-score$ | $A_{cert.}$ [%] | $A_{str.\ attack}$ [%] | $H-score$ |
| ce | 78.41 | 71.25 | 6.81 | 17.16 | 39.68 | 18.23 | 32.06 | 50.72 | 5.44 |
| ceN | **91.65** | 86.26 | 6.62 | **92.42** | 91.41 | 18.72 | 78.42 | 69.01 | 4.82 |
| ceA | 87.55 | 85.05 | 6.84 | 53.11 | 90.82 | 18.76 | 56.72 | 70.36 | 5.33 |
| feD | 88.29 | 86.17 | 6.69 | 89.63 | 91.39 | 19.25 | 64.58 | 71.49 | 5.16 |
| feA | 87.91 | 86.13 | 6.81 | 52.91 | 91.55 | 18.82 | 60.59 | 72.56 | 5.28 |
| llc | 76.00 | 71.50 | 4.94 | 90.96 | 90.27 | 17.47 | 62.44 | 60.79 | 3.62 |
| con | 69.91 | 66.73 | 7.78 | 8.81 | 26.64 | 21.90 | 31.43 | 51.84 | 7.31 |
| conN | 88.49 | 83.00 | 7.63 | 92.13 | 90.39 | **27.06** | **79.41** | 69.10 | 6.56 |
| conA | 85.27 | 82.16 | 7.61 | 11.88 | 90.49 | 4.88 | 51.29 | 71.44 | **9.55** |
| conF | 86.55 | **87.20** | **8.06** | 81.78 | **91.88** | 21.98 | 45.46 | **76.47** | 7.29 |

Table 10: Accuracy of source models (before target retraining) on the target dataset versus accuracy on the source dataset.

| | SVHN → MNIST | | EMNIST → KMNIST | | CIFAR10 → FMNIST | |
|---|---|---|---|---|---|---|
| Train | $A_{Source}$ [%] | $A_{Target}$ [%] | $A_{Source}$ [%] | $A_{Target}$ [%] | $A_{Source}$ [%] | $A_{Target}$ [%] |
| ce | 94.03 | 19.81 | 92.12 | 1.56 | 85.70 | 5.44 |
| ceN | 93.40 | 22.13 | 92.57 | **2.01** | 79.84 | 5.96 |
| ceA | 94.20 | 19.96 | 92.40 | 1.69 | 84.83 | **11.52** |
| feD | 93.69 | 16.79 | 92.29 | 1.87 | 83.46 | 6.39 |
| feA | 94.33 | 17.81 | 92.91 | 1.67 | 84.35 | 3.89 |
| llc | 77.60 | 58.32 | 91.06 | 1.85 | 65.75 | 6.79 |
| con | 93.32 | 65.24 | 92.63 | 1.23 | 87.26 | 9.91 |
| conN | 91.70 | 65.78 | 92.39 | 0.95 | 83.59 | 9.77 |
| conA | 92.41 | **66.17** | 92.34 | 1.08 | 85.39 | 9.96 |
| conF | **95.28** | 25.10 | **93.26** | 1.68 | **89.91** | 10.20 |

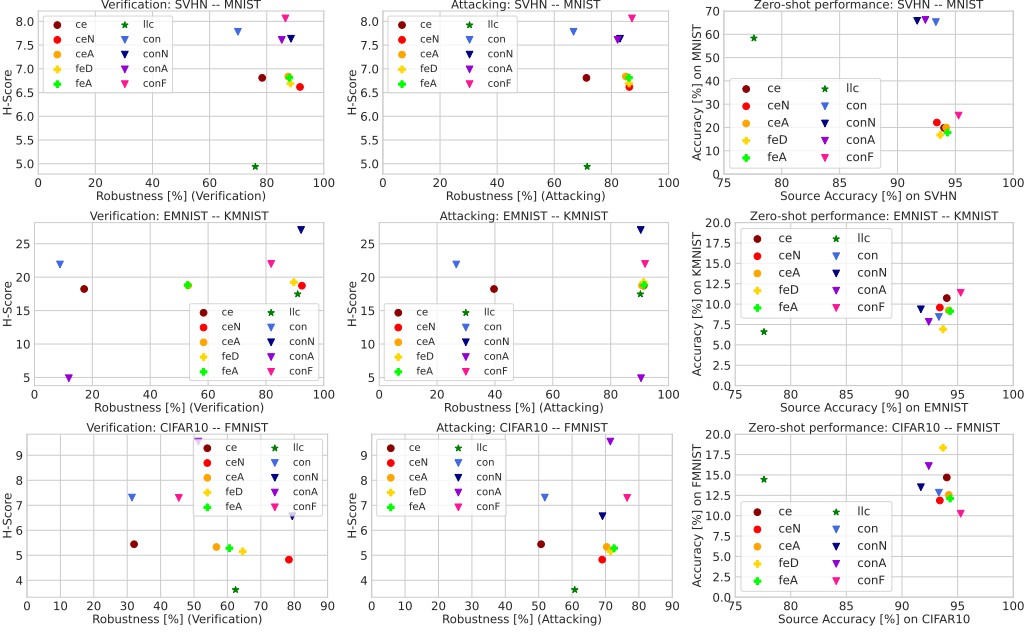

Figure 13: Transferability quantified by the H-Score versus robustness of the source models quantified as verifiable accuracy (Verification) or accuracy under the strongest (FGSM, PGD, DeepFool) attack (Attacking) and zero-shot performance on SVHN – MNIST, EMNIST – KMNIST and CIFAR10 – FMNIST.

