# OpenReview forum: "When is Adversarial Robustness Transferable?"
_ICLR.cc/2023/Conference — Submitted to ICLR 2023_

### Official Review · Reviewer_fomQ · 2022-10-19

**Confidence:** 3
**Correctness:** 3
**Technical Novelty And Significance:** 1
**Empirical Novelty And Significance:** 3
**Recommendation:** 3

**Clarity, Quality, Novelty And Reproducibility:**

Clarity: I think that the writing is clear, but I think there can be some improvements in presentation. First, it would be helpful if the authors bolded the main takeaways from each section of the experiments.  It would also help if the color scheme used is consistent across graphs with similar content (ie. the 2 rows of Figure 4).

Quality: My main concern is that while this paper claims to study transferability of adversarial robustness, the paper only studies L2 adversarial robustness and the size of the perturbation used is much smaller than commonly studied in adversarial ML literature ($\epsilon=0.1$ vs $\epsilon=0.5$).

Novelty: While the methods used in this paper aren't novel, I think there is some novelty in the experiments performed.

**Strength And Weaknesses:**

Strengths:
- extensive experiments: the authors compare many different training techniques on a variety of datasets (SVHN, MNIST, CIFAR-10)
- writing is clear

Weaknesses:
- When comparing verifiable robustness on the source, it would also be informative to provide results with all training techniques augmented with Gaussian noise.  If the base classifier is not robust to Gaussian noise, then it's not surprising that verified robustness via randomized smoothing would be low as randomized smoothing requires that the base classifier is robust to Gaussian noise, which makes the results in the top row of Figure 2 not very informative.
- There are only results for L2 adversarial robustness with $\epsilon= 0.1$.  Commonly in adversarial ML literature $\epsilon=0.5$ with CIFAR-10 is used so I think some results should be provided for this larger radius as well.  Currently the standard trained source model does not drop to close to 0% accuracy with this small perturbation.  Additionally, it would be interesting to see some results for empirical Linf adversarial robustness as well.
- It would also be interesting to see results with stronger attack methods (specifically using either fab-t or apgd-t from AutoAttack) to better measure empirical robustness.

**Summary Of The Paper:**

The authors compare multiple training procedures in order to understand which training techniques improve target robustness.  They find that generally increasing robustness on the source increases robustness on the target.  They also find that techniques involving local lipschitz and contrastive learning generally transfer better.  The authors also find that target retraining doesn't impact target model robustness much.

**Summary Of The Review:**

Overall, I find that this is a well-written paper with extensive experimentation.  However, I think that since the paper claims to analyze the transferability of adversarial robustness, the authors should consider both L2 and Linf attacks with perturbation sizes that align with adversarial ML literature.

---

> ### Comment · Reviewer_fomQ · 2022-12-07
> **Thank you for the general response**
>
> Thank you for the additional clarifications.  I think that additional experimental results (specifically using larger perturbation radius for L2 attacks and considering Linf robustness) are necessary.  I also think that when considering verified robustness via randomized smoothing, it is important to train with data augmented with Gaussian noise that way the underlying classifier is robust to Gaussian noise (which is needed for randomized smoothing).  Because of this, I will leave my score as is.

---

### Official Review · Reviewer_nHzp · 2022-10-21

**Confidence:** 4
**Clarity, Quality, Novelty And Reproducibility:** The paper is easy to follow but the c…
**Correctness:** 3
**Technical Novelty And Significance:** 1
**Empirical Novelty And Significance:** 2
**Recommendation:** 3

**Strength And Weaknesses:**

Strengths:
- The paper makes an experimental exploration on the transferability in adversarial robustness across domains. The topic itself is interesting.
- This paper is well written and easy to follow.

Weaknesses:
- This is a paper with experimental studies and findings. My main concern is that the conclusions are trivial, highly expected with weak evaluations. Specifically
   1. 'conF is the best training procedure.....' is expected as 'distance' loss improves robustness. The authors are suggested to include [1] TRADES as a training procedure. To demonstrate the effectiveness of contrastive method, a comparison between TRADES and TRADES+con is recommended.
   2. 'contrastive learning and llc achieve better robustness during target retraining': The authors are suggested to study on harder datasets (i.e., ImageNet, CIFAR100), larger architectures (i.e., WRN), and stronger attacks (i.e., [2]AutoAttack). The weak evaluation may provide a false sense of robustness.
   3. 'robustness to attacks infer robustness against distribution shifts': sec.4.4 only compares clean test accuracy vs. distribution shift. IMO, a direct comparison between robustness and distribution shift is better.
   4. 'transferability depends on how related between tasks': this finding is trivial as it has been discovered by many previous works.
- Most of contents are covered by [Shafahi et al. (2020)], which uses harder datasets (e.g., ImageNet->Cifar-100, Cifar-100 -> Cifar-10), larger architectures (ResNet50 and WRN32-10), and normal attack settings. Their conclusions are more thorough towards adversarial robustness. The contributions towards verification and robustness to distribution shift, as claimed by authors, are trivial.


**Summary Of The Paper:**

This paper studies how and when adversarial robustness to be preserved and transferred across domains. In experiments, the paper shows that 1) training procedures affect robustness, 2) contrastive learning and llc are more generic thus achieve better robustness during target retraining, 3) training procedure on the source domain has a major effect on target robustness than target retraining, 4) robustness to attacks infer robustness against distribution shifts, and 5) transferability depends on how related between the source and the target domain. However, these findings/conclusions are either trivial or from non-thorough analysis, which significantly reducing the paper contribution.

**Summary Of The Review:**

This paper focuses on the transferability in adversarial robustness across domains. It gives very detailed experimental results.

But it is less likely to be accepted as an ICLR paper. There is no novel idea proposed, though a very detailed experiment process is presented. The evaluations are weak, and the discussions are trivial.

---

### Official Review · Reviewer_zUPp · 2022-10-24

**Confidence:** 4
**Correctness:** 3
**Technical Novelty And Significance:** 2
**Empirical Novelty And Significance:** 1
**Recommendation:** 5

**Clarity, Quality, Novelty And Reproducibility:**

W1. The paper failed to reproduce a common transfer-learning scenario where the source data is much larger than the target data, making the results less useful.
W2. There are no new findings from the experimental results. Another paper “When Does Contrastive Learning Preserve Adversarial Robustness from Pretraining to Fine Tuning” already showed that pre-training in contrastive learning can improve cross-task robustness transferability.
W3. There is no new idea in this paper. All training procedures and target retraining techniques are existing ones. This is not a fatal issue, though, as the paper is on experiments.



**Strength And Weaknesses:**

S1. The authors conduct a wide range of experiments to compare how different training procedures/target retraining techniques affect the adversarial robustness of a model.
S2. The results showing that improving model robustness on the source domain increases robustness on the target domain, while target retraining has a minor influence on target model robustness, are intuitive yet important to the practitioners in the field.


**Summary Of The Paper:**

This work analyzes how different training procedures on the source domain and fine-tuning strategies on the target domain affect model robustness. The authors show that the training procedure on the source domain has a major effect on target model robustness while target retraining has a minor effect. The authors indicate that contrastive learning and training with a local Lipschitz constant best preserve robustness during target retraining.

**Summary Of The Review:**

The authors use a popular transfer learning framework consisting of two parts, a feature extractor f which extracts representations from the inputs and is trained on the source domain and a classifier h which maps extracted representations to predictions and is retrained on the target domain. And the authors investigate and compare how different training procedures and target retraining techniques affect performance and robustness of this model. If the experiments were properly configured, the results would help us understand how transferable is the adversarial robustness after fine-tuning.

Unfortunately, the experiments miss a critical point: in many transfer learning tasks, the source model is likely to be pre-trained on a much larger dataset. This is especially true in the setting of contrastive learning since no human annotated labels are required in the source task. However, the author failed to consider this situation in their experiments, and in all tasks, the source dataset is not significantly larger than the target dataset. This significantly degrade the impact of their results.

Furthermore, the authors of the paper “When Does Contrastive Learning Preserve Adversarial Robustness from Pretraining to Fine Tuning” already showed that pre-training in contrastive learning can improve cross-task robustness transferability. The authors should better position their work and describe the value added.

---

### Official Review · Reviewer_3GPh · 2022-10-24

**Confidence:** 5
**Correctness:** 3
**Technical Novelty And Significance:** 2
**Empirical Novelty And Significance:** 2
**Recommendation:** 3

**Clarity, Quality, Novelty And Reproducibility:**

- **Clarity**: In general, the paper is well written and easy to follow.
- **Quality**: In my opinion the quality of the paper could be improved. As mentioned in **Weaknesses** the experiments are limited to small-scale pretraining tasks, do not test the relevant robustness regimes, and focus only on partially relevant research questions.
- **Novelty**: As far as I know, the methods, results and questions presented in this paper are novel and have not been published in prior work.
- **Reproducibility**: The empirical results of this paper are probably reproducible.

**Strength And Weaknesses:**

# Strengths

1. **Interesting scientific question**: I find, the main premise for this experimental study very relevant to the community. Indeed, understanding when and how does robustness transfer is a problem of great interest.
2. **Evaluation of many robust pretraining baselines**: It is quite admirable that the authors tested and evaluated all their experiments on 10 different pretraining strategies both in terms of empirical accuracy and certifiable robustness.
3. **Clear writing**: Overall, the paper is easy to read.

# Weaknesses
1. **Small-scale pretraining tasks**: Although I can understand that the computational complexity of repeating all the experiments presented in this work using larger-scale datasets for pretraining, such as ImageNet, would be much much higher, it is not very realistic to expect that the same phenomena observed when pretraining on EMNIST or CIFAR10 would be directly observed on ImageNet. This is clearly a very strong weakness of this paper, because even in the case where all the experiments had been flawlessly executed, they will probably show irrelevant results that do not capture the behaviors that are seen in practice (and that serve as motivation in the introduction).
2. **Only very small values of $\epsilon$ are studied and for the $\ell_2$-norm**: Similarly, another important weakness of this work is that it only shows results with very small values of $\epsilon$ (i.e., robustness budget) and only $\ell_2$-norm. $\epsilon=0.1$ is much smaller than the standard values used by the community to evaluate robustness on these datasets (e.g., $\epsilon=0.5$ on CIFAR10 in RobustBench). This is not a small detail, but a very important factor that qualitatively affects the behaviour of the models. For example, at very small values of $\epsilon$ (like the ones used in this paper) it is known that augmenting the training set with random noise can confer some non-trivial accuracy against those weak budgets, however this is clearly not the case at larger $\epsilon$. In this regard, one big thing missing from all the experiments presented in this paper is an ablation study investigating the role of $\epsilon$ and $\ell_p$ norm on the results. And if a proper ablation study is not viable, at least all the experiments should have been evaluated in a much more challenging robustness regime (i.e., larger $\epsilon$ and $\ell_\infty$).
3. **Adversarial robustness is not necessarily connected to robustness to distribution shifts**: In an attempt to increase the breadth of the study, this paper also tries to address the problem of robustness against distribution shifts or common corruptions using the same experimental protocol (i.e., using adversarially pretrained models and fine-tuning on a different task). However, in general, there is still no consensus in the community on whether adversarial robustness and robustness to distribution shifts are actually connected (Yin et al. 2019, Kireev et al. 2022). In this regard, by evaluating only the differences in robustness to distribution shifts of models pretrained to be robust against adversarial perturbations, the paper is focusing again on an irrelevant problem. For this study to be relevant, the paper should compare the finetuning performance of models pretrained with methods that confer robustness to distribution shifts, and not necessarily only to adversarial perturbations.
4. (Minor) **Strange transferability study**: I find the study about the role of representation transferability in downstream robustness a bit odd. Personally, when I first read the motivation of this research question in the introduction, I thought the main question was understanding how the pretraining task and its transferability downstream, influenced the robustness of the fine-tuned model. However, the study focuses only on studying differences in the quality of the representations, as all the experiments are performed for the same pretraining and downstream tasks.

- Dong Yin, Raphael Gontijo Lopes, Jon Shlens, Ekin Dogus Cubuk, Justin Gilmer. A Fourier perspective on model robustness in computer vision. NeurIPS 2019.
- Klim Kireev, Maksym Andriushchenko, Nicolas Flammarion. On the effectiveness of adversarial training against common corruptions. UAI2022.


**Summary Of The Paper:**

This paper presents an empirical study on how different robust pretraining protocols affect the robustness of different downstream tasks. In this regard, many different pretraining strategies are compared on a few small-scale pretraining-finetuning pairs (e.g., SVHN-> MNIST or CIFAR10->FMNIST) in terms of the $\ell_2$ adversarial robustness (at small $\epsilon$) of the fine-tuned models. Based on this study, the authors claim that the best pretraining protocols are those that achieve a general-level of robustness without overfitting to a particular dataset, and that the fine-tuning protocol does not influence too much the final performance. All experiments are performed evaluating both empirical robustness (using diverse attacks) and certified robustness (based on randomized smoothing). Some results are also provided in terms of robustness to common corruptions.

**Summary Of The Review:**

Unfortunately, I believe that the paper does not meet the bar for acceptance to ICLR. The overall research question of this empirical study is very interesting and relevant for the community, but the execution of the study is suboptimal and does not adress the right subquestions. For being accepted this paper would require to perform a much larger scale evaluation of robustness using larger pretraining tasks and multiple downstream targets. The evaluation of robustness should also be conducted at diverse robustness regimes since these have qualitatively different properties.

---

> ### Comment · Reviewer_3GPh · 2022-11-22
> **Answer to authors**
>
> I thank the authors for taking the time to answer to the feedback of all the reviewers. However, after having read their general message, and the other reviewers’ comments, I stand by my previous assessment and will still recommend that this paper is not accepted.
>
> As I mentioned in my original review, to be useful, this paper would require to perform a much larger scale evaluation of robustness using larger scale pre training and fine tuning tasks. Currently, this paper focuses on the wrong questions and has a suboptimal execution which does not meet the bar for acceptance at ICLR in my humble opinion.

---

### Author Response · Authors · 2022-11-15
**Answer to the reviewers**

Dear reviewers,

Thank you for the feedback and the comments. First, we want to  address the main questions.

**Choice of tasks and architectures**: We compare 10 different training procedures, including one that enforces a local Lipschitz constant (llc) on the model. This llc-training restricted the model architecture to the ones we used, since it cannot handle all layer types that are part of e.g. Resnet50. However, we will run the other 9 models on larger architectures and data sets.

**Choice of norm, perturbation size, ablation studies**: We chose the L2-norm since it is more aligned with randomized smoothing than using the Linf norm. Similarly, we chose eps=0.1 since it is a commonly analyzed value in adversarial robustness studies and randomized smoothing. Please also note that in contrast to some other studies, our data is normalized, i.e. between 0 and 1. However, we will additionally include a larger eps in our evaluation.

**Choice of training procedures**: Since reviewer nHzp did not provide any reference, title or author we are not sure which model is meant by “TRADES”. We assume that “TRADES” refers to [1]. We used 10 different training procedures that cover fundamentally different and popular transfer learning methods and adversarial training techniques, which is already a lot. However, we will consider including TRADES as 11th technique. Thanks for the suggestion.

**Robustness evaluation**: We evaluate robustness based on two complementing approaches. First, non-robustness is quantified and illustrated by three attacks of different strength (FGSM, PGD, DeepFool). Second, robustness is quantified by verification based on randomized smoothing. We included verification, since in the past every defense based on attacks could be broken by other attack strategies, which is not possible by the design of verification techniques. Since our attack results and verification results complement each other (without a large gap), using more/different attacks such as AutoAttack will not change the results.

**Adversarial robustness and distribution shifts**: We agree with Reviewer 3GPh that there is still no consensus in the community on whether adversarial robustness and robustness to distribution shifts are connected. We believe that this is why our experiments can help us get closer toward understanding this connection. In any case, we will add a discussion to clarify the (lack of consensus on the) connection.

**Related work**: Shafai et al. 2020 [2] focused on how the depth of the retrained/ finetuned classifier part of the neural network correlates with robustness and improves robustness against PGD attacks (and few-step CW attacks) by using specific end-to-end fine tuning.
Fan et al. 2021 [3] proposes an unsupervised contrastive learning model that improves adversarial robustness against PGD and Autograd attacks.

None of these works compares fundamentally different training strategies such as using a (local) Lipschitz constant, adversarial training and contrastive learning to analyze which one results in the most robust source and target models. Furthermore, for a thorough robustness analysis you need both: attacks of different strength to quantify non-robustness and a complementing verification technique to quantify model robustness.

Moreover, as we point out in the paper the findings in the literature are not comparable with each other, and even lead to contradictory conclusions. While Salman et al. 2020 [4] shows that robust training improves the accuracy on the unperturbed target domain data, Shafahi et al. 2020 [2] shows the opposite – adversarial training increases robustness but decreases accuracy. This is one motivation for our work.

[1] Hongyang Zhang, Yaodong Yu, Jiantao Jiao, Eric P. Xing, Laurent El Ghaoui, Michael I. Jordan, “Theoretically Principled Trade-off between Robustness and Accuracy”, 2019

[2] Ali Shafahi, Parsa Saadatpanah, Chen Zhu, Amin Ghiasi, Christoph Studer, David Jacobs, Tom Goldstein, “Adversarially robust transfer learning”, ICLR 2020.

[3] Lijie Fan, Sijia Liu, Pin-Yu Chen, Gaoyuan Zhang, Chuang Gan, “When Does Contrastive Learning Preserve Adversarial Robustness from Pretraining to Finetuning?”, Neurips 2021.

[4] Hadi Salman, Andrew Ilyas, Logan Engstrom, Ashish Kapoor, and Aleksander Madry. Do adversarially robust imagenet models transfer better?, NeurIPS 2020

---

> ### Comment · Reviewer_3GPh · 2022-11-16
> **New revision?**
>
> Thank you very much for this general comment. Do you plan to upload a revision with the promised changes?

---

> > ### Author Response · Authors · 2022-11-17
> > **New revision**
> >
> > Training, retraining and analyzing 10 different models with the desired architectures on large data sets, adding further training procedures, and further eps values requires time. We started the experiments and if they finish in time we will update the paper, but finishing them before the deadline is unlikely based on our calculations.

---

### Decision · Program_Chairs · 2023-01-20

**Decision:**

Reject

**Justification For Why Not Higher Score:**

Reviewers pointed out important missing experiments, e.g., L2 and Linf attacks with perturbation sizes, source model pre-trained on a much larger dataset, etc. Also, the fact that pre-training in contrastive learning can improve cross-task robustness transferability has already been studied before, and the authors need to better clarify the difference between this work and the literature.

**Justification For Why Not Lower Score:**

N/A

**Metareview: Summary, Strengths And Weaknesses:**

This paper has done an empirical study to explain when adversarial robustness is transferrable. Though this is an interesting problem, it is a pity that none of the reviewers would like to accept this paper, because of quite a lot of major concerns.